# Knowledge graph-based recommendation framework identifies drivers of resistance in EGFR mutant non-small cell lung cancer

Anna Gogleva[1], Dimitris Polychronopoulos [2], Matthias Pfeifer[3], Vladimir Poroshin[4], Michaël Ughetto[5], Matthew J. Martin[3], Hannah Thorpe[3], Aurelie Bornot[6], Paul D. Smith [3], Ben Sidders[2], Jonathan R. Dry [7], Miika Ahdesmäki[2], Ultan McDermott [3], Eliseo Papa[1✉] & Krishna C. Bulusu [2✉]

Resistance to EGFR inhibitors (EGFRi) presents a major obstacle in treating non-small cell lung cancer (NSCLC). One of the most exciting new ways to find potential resistance markers involves running functional genetic screens, such as CRISPR, followed by manual triage of significantly enriched genes. This triage process to identify 'high value' hits resulting from the CRISPR screen involves manual curation that requires specialized knowledge and can take even experts several months to comprehensively complete. To find key drivers of resistance faster we build a recommendation system on top of a heterogeneous biomedical knowledge graph integrating pre-clinical, clinical, and literature evidence. The recommender system ranks genes based on trade-offs between diverse types of evidence linking them to potential mechanisms of EGFRi resistance. This unbiased approach identifies 57 resistance markers from >3,000 genes, reducing hit identification time from months to minutes. In addition to reproducing known resistance markers, our method identifies previously unexplored resistance mechanisms that we prospectively validate.

[1] Biological Insight Knowledge Graph (BIKG), AI Engineering, R&D IT, AstraZeneca, Cambridge, UK. [2] Early Computational Oncology, Research and Early Development, Oncology R&D, AstraZeneca, Cambridge, UK. [3] Bioscience, Research and Early Development, Oncology R&D, AstraZeneca, Cambridge, UK. [4] NLP Lab, Enterprise AI Services, IGNITE, AstraZeneca, Cambridge, UK. [5] Biological Insight Knowledge Graph (BIKG), AI Engineering, R&D IT, AstraZeneca, Gothenburg, Sweden. [6] Data Sciences & Quantitative Biology, Discovery Sciences, R&D, AstraZeneca, Cambridge, UK. [7] Early Computational Oncology, Research and Early Development, Oncology R&D, AstraZeneca, Waltham, MA, USA. ✉email: elipapa@alum.mit.edu; krishna.bulusu@astrazeneca.com

In this study we explore how a biological question can be translated into a recommendation problem[1]. Traditionally recommendation systems have been used to help users discover relevant items among an overwhelming number of options[2]. By interacting with recommendations users provide either implicit or explicit feedback, which recommendation models use to personalize and improve predictions[3]. Information overload is particularly common in e-commerce[4], streaming[5] and social media applications[6], hence recommendation systems play key role in these industries. The biomedical domain on the other hand, is not seen as a typical application area for recommendation systems. Their usage so far is limited to a handful of recent studies: Ozsoy et al., applied collaborative filtering for drug repositioning problem[7]; Frainay et al., developed a network-based recommendation solution to enrich and interpret metabolomic data[8]; Suphavilai et al., built a matrix factorization-based recommendation system to predict response of cancer drugs[9]; Radivojevic et al., developed an automated recommendation solution for synthetic biology[10]. The success of these pilot studies suggest there are potential opportunities for recommendation systems in the biomedical domain; the amount of biomedical data is growing exponentially and scientists could benefit from recommendation solutions that help them to navigate the data and reason about it.

Naturally, direct transfer of classic recommendation approaches to the biomedical domain is not trivial. Specifics of the problem space impose numerous challenges for a recommendation system practitioner, to name a few:

- an elementary unit of recommendation is not a simple self-contained item (e.g a gene), but rather a research direction accompanied by a biologically sound hypothesis;
- ultimate validation of recommendations is complex and often requires expensive and time-consuming laboratory experiments, as opposed to users just "selecting" an item in a common non-biological recommendation scenario;
- unlike traditional applications, in a biomedical setting both implicit and explicit feedback is scarce, making it harder to tune and train models;
- ground truths are scarce and in most cases context-dependent, which renders training challenging;
- due to the high cost associated with accepting a recommendation, an increased emphasis is placed on explainability and exposing causal reasoning paths behind a recommendation.

Despite these challenges, wider adoption of recommendation approaches holds plenty of opportunities to support and accelerate biological research. To illustrate this point, in this study we focused on the problem of drug resistance in lung cancer. Our goal was to build a recommendation solution that finds key genes driving drug resistance. Similar problems are also often formulated as gene prioritization tasks and have been previously addressed with network-based methods[11,12], kernel-based learning[13], and most recently—deep learning approaches[14,15], to name a few. In this study we were interested to explore the lung cancer resistance problem through the lens of recommendation approach.

Drug resistance is a complex biological phenomenon that hinders development of efficient and lasting cancer treatments[16]. Tumors recruit diverse strategies to escape selective pressure induced by drugs, such as changes in drug metabolism[17], inhibition of cell death[18], epigenetic alterations[19] or acquired mutations in drug targets[20]. Enhanced DNA repair and increased amplification of tumor driver genes also contribute to secondary resistance[21]. Genetic and epidemiological diversity of patients[22] further complicates the resistance landscape.

In this study we focus on non-small cell lung cancer (NSCLC) carrying activating mutations of the epidermal growth factor receptor (EGFR). It accounts for 15-20% of lung cancer patients[23]. Treatment with first or second generation EGFR tyrosine kinase inhibitors such as gefitinib, erlotinib or afatinib results in impressive response rates in patients initially[24], however, tumors quickly develop resistance to treatment. The majority of resistant cases are driven by accumulation of secondary mutations of EGFR gene, such as T790M, that prevent binding of EGFR TKI (tyrosine kinase inhibitors) compounds[25]. Development of osimertinib, a third generation EGFR TKI, provided the ability to target such secondary EGFR mutations[26]. In fact, treatment with osimertinib significantly improved patient survival in first-line therapy setting[27]. However, therapy resistance prevails. Acquired mutations of EGFR such as C797S drive osimertinib resistance in 6–26 % of cases. Bypass pathway activation, amplifications of MET or mutations in PIK3CA have also been shown to contribute to resistance[28]. Still, in half of the cases the molecular resistance mechanisms remain unknown and promising markers could reside in a so called "dark matter" of the human genome[29].

A common strategy to find key drivers of acquired resistance is based on functional genomic screens, such as CRISPR screens[30]. CRISPR-Cas9 genome-wide knock out, knock down and knock-in screens have recently emerged as an efficient high-throughput technology to systematically investigate resistance mechanisms[30]. CRISPR screens can be applied in two ways to understand drug response and drug resistance. First, they can be used to identify alterations in genes that increase sensitivity of a cell to drug treatment. Here, researchers measure negative selection of modified genes in drug treatment. This approach can help to define therapeutic combinations that might increase response to treatment. Second, CRISPR screens are used to identify genes that drive drug resistance if altered. In this case the experimental set-up mimics treatment scenarios in the clinic. In this approach, outgrowth or positive selection of drug resistant cells is measured and used to define mechanisms that drive resistance. These can be potentially targeted once resistance is established.

In these settings, a typical output of a CRISPR screen may identify many hundreds of resistance genes. To narrow down the list to the most promising, biologically plausible and actionable resistance genes, researchers have to perform manual triage and validation. During this process experts aggregate prior knowledge about a disease with additional evidence available from clinical and pre-clinical studies and decide which genes to prioritize for experimental validation. The selection process is tedious and time consuming. It also relies on deep specialized knowledge, hence the results can be prone to the individual bias. Our goal was to replace such manual triage with a recommendation solution, which could efficiently integrate diverse types of evidence and identify the most promising candidate genes driving drug resistance.

By moving the problem to a recommendation domain we encounter two major challenges. First is the lack of training data. Here we are dealing with a highly specific molecular phenotype of a poorly understood origin, which prevents us from using information on resistance markers relevant for other, even closely related, diseases as training data. Second, unlike a typical recommendation scenario, in our case both explicit and implicit feedback are lacking. This fact limits our ability to gradually train and improve models. Given these constraints we followed an unsupervised recommendation approach, which relies on content-based filtering. We formalized re-ranking of CRISPR hits as a multi-objective optimization problem[31], where diverse and conflicting types of evidence supporting gene's relevance are

mapped to objectives. During the optimization procedure feasible solutions (genes) are identified and compared until no better can be found. A crucial component of such framework is a set of hybrid features. Each feature represents a distinct type of evidence, such as literature support, clinical and pre-clinical evidence.

Along with the purely biological features, our recommendation system relied on data derived from a specially constructed heterogeneous biomedical knowledge graph[32]. Knowledge graphs provide a convenient conceptual representation of relationships (edges) between entities (nodes). In the recommendation context knowledge graphs gain popularity as a way to introduce content-specific information and also to provide explanations for the resulting recommendations[33]. In addition, graph-based recommendations were shown to achieve higher precision and accuracy compared to alternative approaches[34–36]. We used graph structural information together with graph-based representations to express relevance of a gene in the resistance context. Our assumption was that by combining graph-derived features with clinical ones we could discover unobvious genes that drive drug resistance in lung cancer.

In summary, in this study we explored how a question of finding drivers of secondary EGFR TKI resistance could be addressed as a recommendation problem. We demonstrate that a recommendation system based on multi-objective optimization approach can be used to re-rank CRISPR hits in the context of secondary drug resistance. The proposed framework, together with an automated feature generation flow and interactive re-ranking interface, helped to reduce gene hit prioritization time from months to a few minutes.

## Results

### Re-ranking of CRISPR hits can be approached as multi-objective optimization

We framed re-ranking of CRISPR hits as a multi-objective optimization problem. In this setting, diverse lines of evidence that support gene's relevance are treated as multiple objectives (Fig. 1). In other words, the formal goal is to simultaneously optimize $k$ objectives, reflected in $k$ objective functions: $f_1(x), f_2(x), \ldots, f_k(x)$. Individual functions form a vector function $F(x)$:

$$F(x) = [f_1(x), f_2(x), \ldots, f_k(x)]^T \quad (1)$$

where $x = [x_1, x_2, \ldots, x_m] \in \Omega$; $x$ represents the decision variable, $\Omega$-decision variable space. Therefore, multi-objective optimization can be defined as minimization (or maximization) of the objective function set $F(x)$. With multiple competing objectives a singular best solution often cannot be found. However, one can identify a set of optimal solutions based on the notion of Pareto dominance[37]. A solution $x_1$ dominates solution $x_2$ if the following two conditions are true:

- solution $x_1$ is not worse than $x_2$ according to all objectives;
- solution $x_1$ is strictly better than solution $x_2$ according to at least one objective.

If both conditions are true, we can say that $x_1$ dominates $x_2$, which is equal to $x_2$ being dominated by $x_1$. In other words, dominant solutions can not be improved any further without compromising at least one of the other objectives. A set of such dominant solutions forms a Pareto front, which combines the best trade-off combinations between competing objectives. Therefore, by computing Pareto fronts on diverse sets of objectives defined based on CRISPR screen data and additional supporting evidence we can narrow down the number of promising markers of EGFR TKI resistance (Fig. 1).

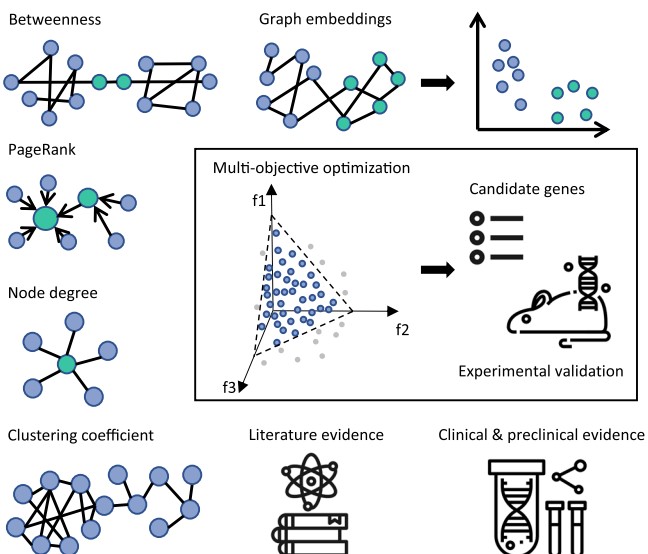

**Fig. 1 Recommendation system takes into account diverse types of evidence to suggest promising drivers of drug resistance in NSCLC.** The evidence from specially built knowledge graph, literature, clinical and pre-clinical datasets is aggregated and formalized as objectives in multi-objective optimization (MOO) task. Recommended solutions (genes) represent the optimal trade-offs between the conflicting objectives. A subset of recommended genes is passed for the experimental validation.

**A hybrid set of features supports recommendation system**. To support the recommendation system we assembled a hybrid set of rich features (Fig. 1 and Supplementary Table 1), with an idea that each feature represents an objective. The selected features were relevant for EGFR inhibitor resistance in NSCLC and corresponded to distinct lines of evidence. Key feature types and rationale to consider them for re-ranking of CRISPR hits are summarized below.

*CRISPR.* CRISPR screen data served as a starting point for re-ranking. In this study we relied on screens that were set-up to resemble clinical treatment scenarios for *EGFR* mutant lung cancer, using NSCLC cancer cell lines harboring *EGFR* mutations commonly found in patient populations and where those cell lines were treated with 1st or 3rd generation EGFR inhibitors. In total we identified a starting list of 1550 candidate drug resistance genes[38] that were labeled as significant after the screen analysis. We further aggregated CRISPR data by computing consistency metrics, which reflected stability of a gene's performance across experimental conditions. Normally genes showing consistent behavior in multiple relevant conditions, e.g related cell lines or treatments, are ranked higher by domain experts. Altogether, seven consistency-based features were incorporated in the feature set: (1) three features based on loss-of-function part of the screen; (2) three features based on gain-of-function part of the screen; (3) a summary metric reflecting overall consistency in the full screen (Supplementary Table 1).

*Literature-based metrics.* Literature search is routinely used as a first step to confirm experimental findings and to find support for a potential mechanistic hypothesis. For the EGFR inhibitor resistance problem we were primarily interested in the overall literature support for a gene. As a proxy of literature support we calculated the total number of publications that mention a gene in a relevant context, such as "cancer", "resistance", "EGFR", "NSCLC". Conveniently, the same exact metric when reversed can be interpreted as novelty of a particular target. To extract

literature mentions we analyzed a total corpus of >180,000 PubMed papers published between 2000 and 2019. We included aggregated literature metrics, based on two terms of interest: EGFR and NSCLC. For each gene we computed the number of papers that mention a gene together with one of these terms (Supplementary Table 1). To account to the fact that mentions in research papers vary drastically between genes, we also included normalized literature-based frequencies.

*Graph-derived features.* In this study we used a custom knowledge graph (KG) as a source of side information for the recommendation system. Our KG contained 11 million nodes and 84 million edges and was composed of 37 public and internal datasets, such as Hetionet, OpenTargets, ChEMBL and Ensembl[32]. In general, patterns of interactions between biological entities captured by knowledge graphs can be translated into features and consumed by recommendation systems in a number of ways (Fig. 1 and Supplementary Table 1). One way is to compute features directly on the graph. This includes metrics such as node degree—reflecting the importance of a node; PageRank—a measure of node's popularity[39]; betweenness—a way to detect the amount of influence a node has over the flow of information in the graph. An alternative approach involves projecting the graph into a low-dimensional space, so that every node is transformed into its vector representation—embedding. Embeddings capture critical structural properties of the graph[40], so that the nodes that were close in the graph also remain close in the embedding space. In this study we computed distances in the embedding space between each gene and two key entities of interest: "EGFR" and "NSCLC". The assumption is that genes most relevant to the EGFR TKI resistance phenotype should be close to either lung cancer or EGFR gene nodes.

*Clinical enrichment scores.* To ensure the recommendation system captures clinical evidence, we included genomic data from osimertinib-treated EGFR-mutant lung cancer patients in the feature set. We prioritized five clinical trials: AURAext[41], AURA2[42], AURA3, FLAURA[27], and ORCHARD[43]. The prevalence of genomic alterations in non-responders vs. responders across 355 patients treated with osimertinib were calculated and included as "clinical enrichment score features" to the feature set (Supplementary Table 1).

*Tractability and gene essentiality.* Traditionally drug resistance in cancer is addressed by developing compounds or combination therapy that modulates activity of its key driver genes (targets). When a target is prioritized for drug development one needs to ensure that: (1) a gene is tractable in principle, i.e., it is shown or predicted to bind to commonly used drug modalities with high affinity; (2) a gene should not be essential, since knock-out of an essential gene can be detrimental to other cells in the organism, not just the tumor ones. To support the first consideration, we included bucket tractability estimates[44] for three modalities: antibodies, small molecules and other modalities (enzyme, oligonucleotides, etc). In support of the second consideration we integrated DepMap[45] essentiality estimates.

In summary, the final hybrid set contained 27 rich features, supporting diverse criteria taken into consideration during validation of CRISPR hits by domain experts (Supplementary Table 1). The hybrid set was also augmented by graph-derived features and literature-based metrics. Correlation analysis of the hybrid feature space indicated expected patterns: (1) strong positive correlation between structural graph features, such as degree, pagerank and betweenness; (2) negative correlation between CRISPR features derived from knock-out and activation screens (Supplementary Fig. 7).

**Interactive interface allows experts to re-rank CRISPR hits**. So far, we have defined a basic model for multi-objective optimization and demonstrated how to build a hybrid set of features to support re-ranking of CRISPR hits in the EGFR TKI context.

In the real-world scenario, decision-making can be both iterative and subjective. A choice of a particular set of objectives and the direction of optimization for the same variable varies from expert to expert. Each combination of objectives and corresponding directions for optimization might result in a different shape of Pareto front, therefore - in a different set of top recommended genes.

To accommodate diversity of opinions and to enable domain scientists to explore complex trade-offs between the objectives we built an interactive application - SkywalkR https://github.com/AstraZeneca/skywalkR[46] (Fig. 2). SkywalkR is a Shiny app[47], which operates on top of the pre-assembled hybrid feature set (see Supplementary Table 1). SkywalkR app combines diverse facets of knowledge to guide re-prioritization of CRISPR hits for experimental validation. In addition, it allows domain experts to explore various trade-offs between objectives. Thereby, it stimulates exploration of possibilities, highlights gaps in the existing knowledge and motivates to adjust expectations about optimal solutions.

Automated engineering of rich features coupled with multi-objective optimization realized through SkywalkR interactive interface dramatically reduced the time required for gene prioritization from a few weeks to minutes.

**Evaluation demonstrates majority of top recommendations labeled as credible by experts**. To evaluate the recommendation framework against expert opinions we fixed a default set of preferences. Preferences were defined by a combination of selected objectives and corresponding directions of optimization. The set of defaults was chosen to mimic the process of CRISPR hit validation by domain experts, but also included graph-derived features and summary metrics extracted from the literature (Supplementary Table 1). The resulting list contained 57 recommended genes (Fig. 3). To collect opinions on the list from the domain experts, we set-up an interactive evaluation task with Prodigy[48]. Five independent experts assigned each of the recommended genes to one or more predefined categories: (1) known, resistance marker; (2) previously unknown, credible hit; (3) previously unknown hit, unclear tractability; (4) not novel, not credible hit. Here "unclear tractability" referred to the absence of a clear path to biological validation.

Despite the expected discrepancies between the expert opinions, the majority of the recommended genes (86%) were classified as either "previously unknown, credible hit" or "known resistance marker" (Fig. 3). To consolidate opinions we assigned the most frequent label to each gene (best label). This resulted in three major categories: "known", "previously unknown, credible hit" and "previously unknown hit, unclear tractability". To determine underlying data structure that supported separation between the three labels, we analyzed values corresponding to the objectives included in the default preference. For easier comparison values were standardized (Fig. 3). Genes labeled as "previously unknown, unclear tractability" were clearly separated from the remaining genes on the basis of low values of all objectives across the board, except the log fold-change values from the RNA-Seq study 3. This analysis suggests that in general the experts tend to prioritize genes supported by several lines of evidence.

**Shapley values indicate the high impact of CRISPR-derived features**. To further estimate what was the impact of each of the

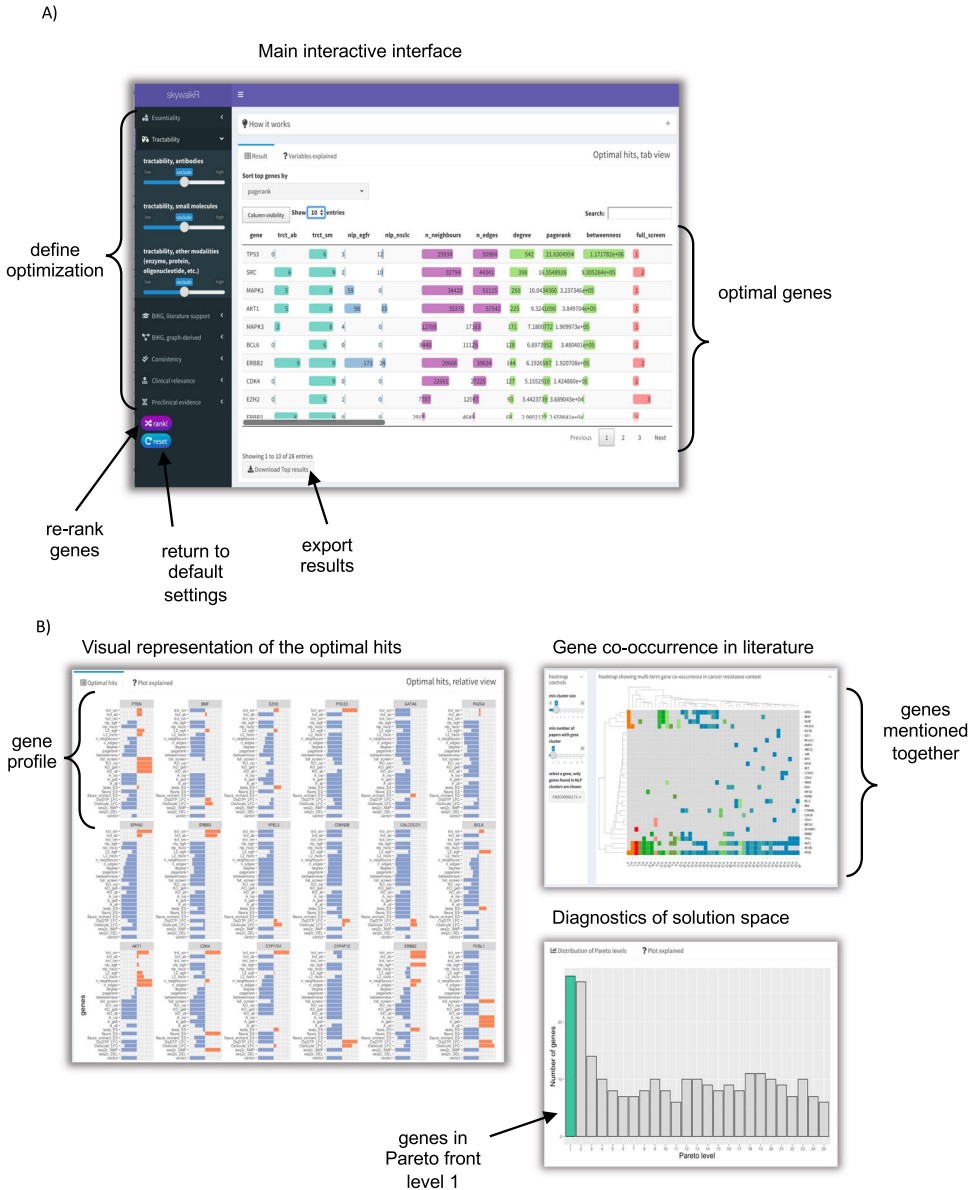

**Fig. 2 SkywalkR interactive interface allows users to re-rank CRISPR hits based on various combinations of objectives. A** On the side bar panel each objective is represented with a slider. Users can decide which objectives to include in the optimization and can also specify direction of optimization (minimize or maximize). **B** Additional tools to explore the results. Relative view shows profiles of recommended genes. Bar plots demonstrate standardized values across objectives and top recommended genes. Co-occurrence heatmap demonstrates clusters of genes frequently mentioned together in EGFR TKI resistance context.

objectives on the expert decisions, we calculated Shapley values[49], a game-theoretic technique used to explain output of machine learning models. For this analysis we reduced the problem to a binary classification task, where a gene is either selected by an expert or not. To assign positive labels we used a set of 100 genes prioritized to a secondary CRISPR screen[38] and trained two random forest models: (1) based on a default subset of features; (2) based on the full set of rich features, including clinical, pre-clinical, literature, CRISPR and graph-derived categories (Fig. 4).

The resulting Shapley values indicate that in general, CRISPR-derived features have the largest impact on gene classification in both experiments. When the default set of features was tested, the composite CRISPR screen consistency variable accounted for the most impact on classification (Fig. 4). Interestingly, in both cases - when we used either full or only default set of features to train the model, CRISPR-derived features remained decisive, followed

by graph-derived features. Contrary to our expectation, clinical, pre-clinical and literature-derived features had less impact on gene classification in this setting. This suggests that graph-derived features though not routinely used in manual triage, could provide a valuable insight in the overall relevance of a gene, especially when combined with context-defining experimental data, such as CRISPR.

**Network analysis and clinical knowledge indicate EGFR resistance mechanisms among top recommended genes**. To link the prioritized hits to known EGFR biology, we performed pathway enrichment analysis. It captured pathways related to resistance such as "mechanisms of resistance to EGFR inhibitors in lung cancer" and "Anti-apoptotic action of ErbB2 in breast cancer" (Supplementary Table 2). Occasionally enrichment results are

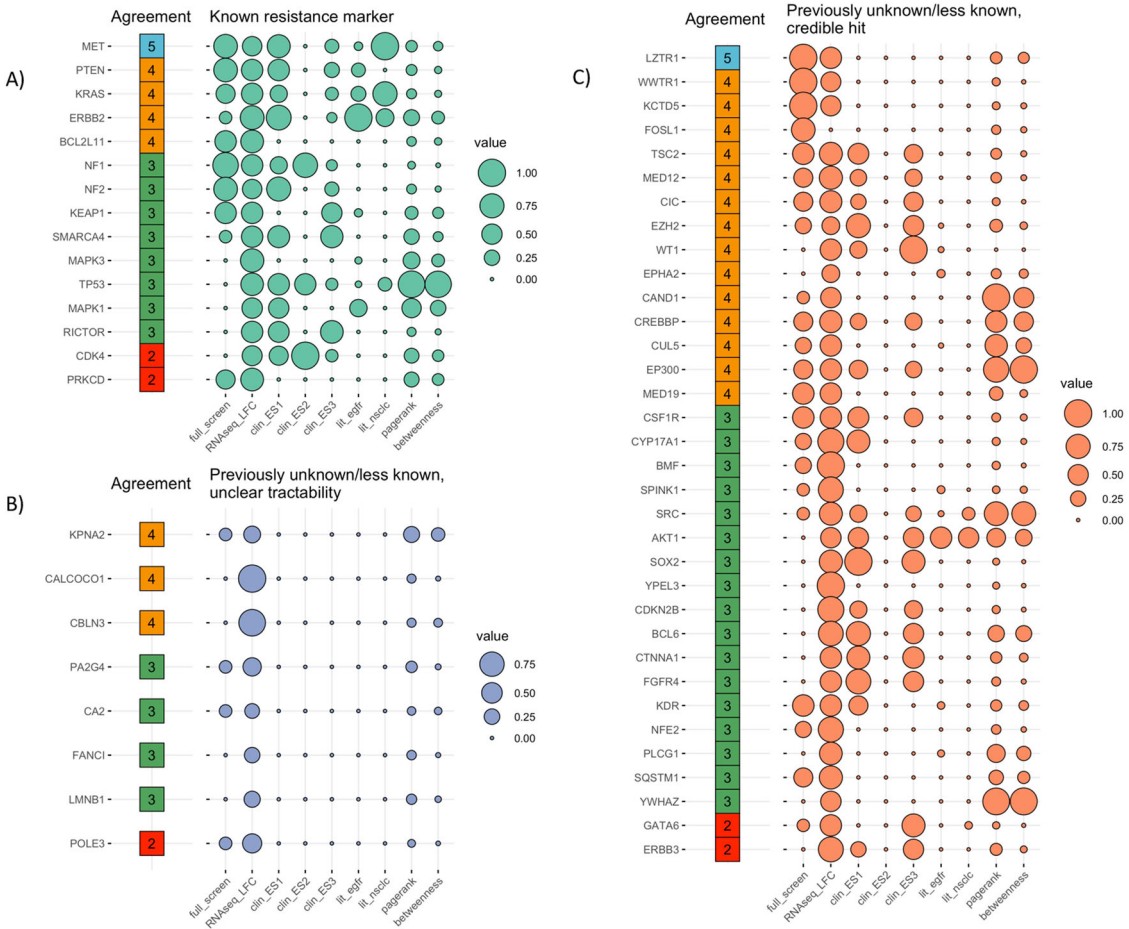

**Fig. 3 Expert curation of recommended hits along with features used for hit triaging.** Evaluation of top recommended genes by five independent experts indicates that majority of genes can be classified either as **A** "known resistance markers" or **C** "previously unknown/less known, credible hits". Eight genes were labeled as "previously unknown/less known hits with unclear tractability" by the experts (**B**). Here, unclear tractability refers to the absence of clear path to validate predictions experimentally. "Previously unknown/less known" is definned within the biological context of this study. Features used to generate predictions: full_screen—overall consistency across all conditions in the CRISPR screen; RNASeq_LFC-log2 fold-change from internal RNASeq study; clinical_ES1, clinical_ES2, clinical_ES3-enrichment scores from clinical studies where resistant patients were compared against responders; lit_EGFR, lit_NSCLC-co-occurrence estimates from the literature; pagerank—"popularity" measure of a node; betweenness—centrality estimate of a node. "Agreement" column indicates the number of experts assigning a certain label to a gene. Size of a bubble reflects value normalized across the full set of features for all genes.

redundant and top terms may be similar to each other thus carrying little new information. Therefore, we also performed crosstalk analysis (Supplementary Fig. 8), which confirmed "mechanisms of resistance to EGFR inhibitors in lung cancer" among the top enriched pathways.

To additionally annotate recommended genes with clinical relevance information, we performed a comparison with OncoKB[50]. OncoKB assesses genetic alterations by annotating each gene under five categories: therapeutic, prognostic, diagnostic, resistance, and FDA-levels. As expected, this assessment captured genes with known clinical significance (Supplementary Table 3).

**Experimental validation demonstrates key regulatory roles for epigenetic and Ras-signaling genes in mediating resistance phenotype.** To further validate a subset of recommended genes we experimentally investigated their direct impact on osimertinib resistance. For experimental validation of our recommendations, we identified biological mechanisms that have recently gained prominence within the domain of EGFRi resistance but do not yet have an approved drug target within the pathway (Hippo pathway—WWTR1 and NF1[51] and KCTD family[52]). In addition,

we selected targets that have inhibitors available to assess combination benefit in resistance models when a recommended target is inhibited in combination with osimertinib (SRC and EZH2). Lastly, to provide ground truth we included established EGFRi resistance markers as the background to our validation studies (MET and PTEN). We manipulated expression of six recommended genes (MET, WWTR1, EZH2, PTEN, NF1, and KCTD5) in *EGFR* mutant NSCLC cell lines sensitive to osimertinib (Fig. 5A). Genes for validation were selected from "known" (as true positives) as well as "previously unknown" categories (Fig. 3).

PTEN, NF1 and KCTD5 were previously shown to negatively regulate MAPK or PI3K/AKT signaling[53–55], known drivers of EGFR TKI resistance. Hence our expectation was that deregulation of PTEN, NF1 and KCTD5 should mediate a stable resistance phenotype. To test this hypothesis we established a flow cytometry-based long-term competition assay (Fig. 5B). The assay showed that after 14 days of co-culture under control (DMSO) condition, perturbation of NF1, PTEN or KCTD5 expression (Fig. 5B, C) did not affect proliferation in comparison to non-targeting control (NTC) cells. However, when treated with osimertinib NF1 or PTEN KO caused a fitness advantage, measured as a two to three fold increase in proliferation (in PC-9 or HCC827, respectively) compared to control cells (Fig. 5C).

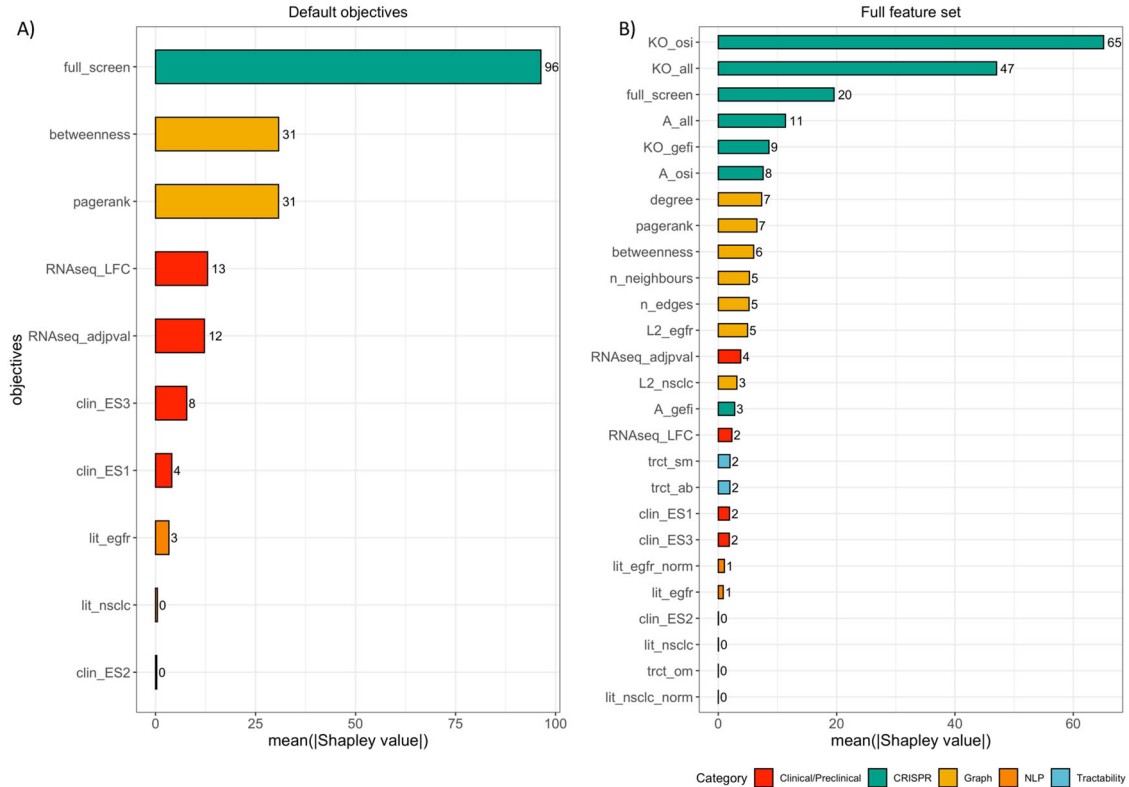

**Fig. 4 Shapley values reflect relative importance of features that differentiate relevant hits from non-relevant.** We modeled the task as a binary classification problem, where a gene can be labeled as either relevant or not. A list of genes prioritized by domain experts for secondary screens was used as a set of labels. The analysis was run using **A** a set of default objectives; **B** similar analysis repeated on the full set of objectives. In both cases features associated with consistency in CRISPR screens appear to have the greatest impact on classification, closely followed by graph-derived features.

In case of KCTD5, KO the observed resistance effect was more pronounced in PC-9 compared to HCC827 cells.

Inhibition of EZH2 expression in lung cancer was recently described as sensitizer to anticancer treatment[56]. We abrogated EZH2 expression in II-18 cells (Supplementary Fig. 9D) and tracked proliferation under control and treatment condition. Loss of EZH2 expression in II-18 induced a pronounced osimertinib resistance phenotype. Regarding MET, recent studies suggest that its amplifications[28] are often associated with an overexpression of receptor tyrosine kinase, resulting in bypassing of EGFR signals[57]. To validate relevance of MET to osimertinib resistance we activated its expression in PC-9 cells (Supplementary Fig. 9B) and compared proliferation in control (DMSO) and drug treatment. As expected, overexpression of MET did not alter cell proliferation under control conditions. In contrast, cell proliferation was substantially elevated in cells under osimertinib treatment, compared to control modified cells (Fig. 5E).

We obtained similar results when we activated expression of WWTR1 in PC-9. WWTR1 is an effector of transcriptional activity in the Hippo pathway. This pathway was recently linked to EGFR inhibitor resistance[58]. As demonstrated in long-term clonogenic assays, WWTR1 activation in PC-9 (Fig. 5F and Supplementary Fig. 9B) resulted in substantial outgrowth of colonies resistant to osimertinib treatment, compared to NTC control cells.

To target recommended resistance hits therapeutically, we combined available inhibitors with EGFR inhibition in osimertinib sensitive and resistant cell lines.

One recommended hit from our analysis was the SRC proto-oncogene, a non-receptor tyrosine kinase, which has been previously implicated in EGFR-TKI resistance[59]. To validate SRC as a resistance driver, we employed our panel of acquired-resistant cell lines (Fig. 6A and Supplementary Fig. 10A, C, G) as previously described[60]. We generated dose–response curves for three small molecule SRC inhibitors (ECF-506, dasatinib or saracatinib), comparing sensitivity in parental and resistant clones. Note that resistant clones were co-treated with osimertinib, in which they readily proliferate. We found that parental cell lines were generally resistant to the monotherapy effects of SRC inhibitor treatment (Figs. 6B and Supplementary Fig. 10B, D–F, H). Critically, treatment with equivalent doses of these SRC inhibitors could sensitize all osimertinib-resistant (OR) PC-9, HCC827, NCI-H1975 and HCC4006 cell lines to the clinically relevant dose of osimertinib (160 nM), thus highlighting the importance of SRC in mediating osimertinib resistance.

Combination of osimertinib with EZH2 inhibitor—tazemetostat at increasing concentrations revealed a dose-dependent increase in resistance to osimertinib. Thus, we could demonstrate by genetic and pharmacological means, that abrogation of EZH2 function can increase osimertinib resistance (Fig. 6C). In summary, we collected initial experimental evidence indicating that manipulation of a subset of recommended genes—EZH2, KCTD5, MET, NF1, PTEN, SRC and WWTR1 causes osimertinib resistance.

In addition to known osimertinib resistance markers, and validated markers discussed above, our method also identified several markers of osimertinib resistance with very limited prior knowledge or literature evidence, but could be exciting opportunities as targets in NSCLC treatment or for defining new drug combinations with osimertinib (Fig. 3). Here, we define "previously unknown" as genes having no publications that directly associate them to EGFR in the context of resistance.

Two of the hits, FOSL1 and BCL6, have since been shown to be involved in key molecular bypass mechanisms for EGFR-TKI

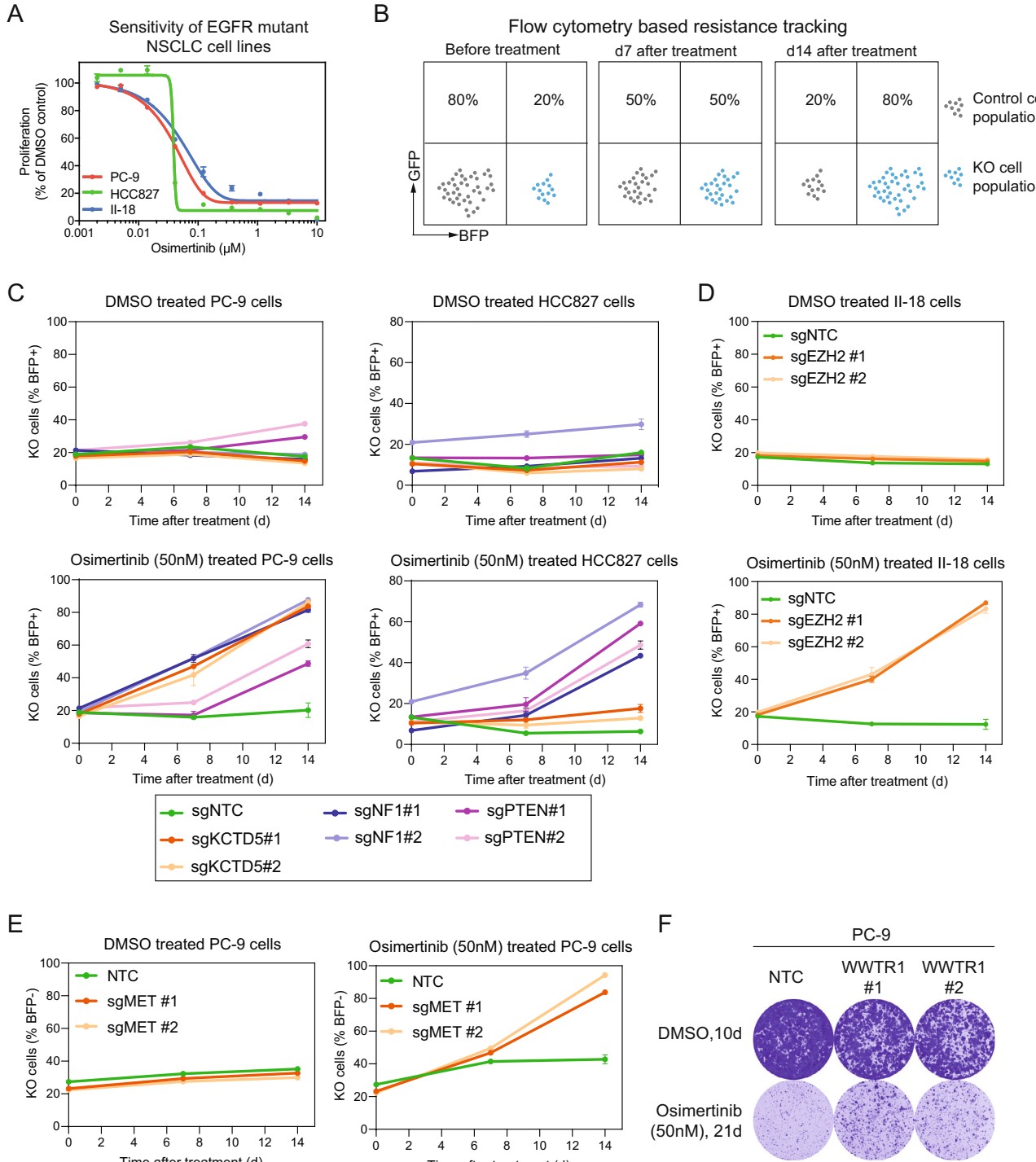

**Fig. 5 Validation of resistance genes proposed by recommendation system. A** Proliferation assays confirm osimertinib sensitivity of EGFR-mutant NSCLC cell lines PC-9, II-18 and HCC827. Data are presented as mean values +/− SD, $n = 2$ technical replicates, representative data of three biological replicates. **B** Resistance to osimertinib by target gene KO was measured in flow cytometry-based competition assays. Increase in percentage of KO cells compared to control cells was considered as resistance effect to osimertinib. **C**, **D** Confirmation of resistance to osimertinib in *EGFR* mutant NSCLC KO cell line models. Effect of KO of genes (EZH2, KCTD5, NF1 and PTEN) on osimertinib resistance was tested in II-18, PC-9 and HCC827 cell lines. Proliferation of cells in DMSO or osimertinib (50 nM) was tracked over 14 days and percentage of BFP+ KO cells was plotted. **E** Activation of MET drives resistance to osimertinib in *EGFR* mutant PC-9. Cells were engineered to express the dCas9-VP64 SAM CRISPR for MET activation. For **C–E**, use of two independent guide RNAs per gene were considered as biological replicates ($n = 2$), three technical replicates are visualized. Data are presented as mean values +/− SD. **F** Activation of WWTR1 drives resistance to osimertinib in *EGFR* mutant PC-9 cells as measured in 21 day clonogenic assays. PC-9 cells were engineered to express the dCas9-VP64 SAM CRISPR for WWTR1 activation. Data presented as mean values +/− SD, representative data of two biological replicates, each consisting of technical duplicates[96].

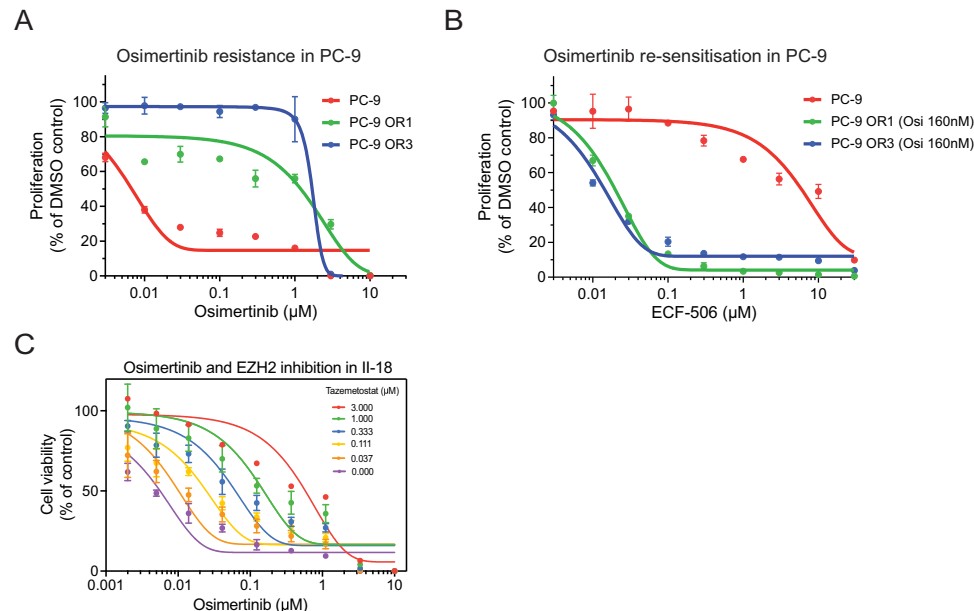

**Fig. 6 Validation of SRC and EZH2 as resistance drivers proposed by recommendation system. A** Osimertinib dose–response curve in PC9 parental cells compared to two lines derived to have acquired osimertinib resistance, as measured by Cell Titer Glo (96h treatment). **B** ECF-506 (SRCi) dose–response curve in PC9 parental vs. resistant lines, as measured by Cell Titer Glo (96 h treatment). Resistant cells were co-treated with 160 nM osimertinib. **C** Viability assay demonstrating dose-dependent effect of EZH2 inhibition on osimertinib resistance in Il-18. OR = osimertinib (Osi) resistant. Data are presented as mean values +/− SD ($n = 3$) of a typical plot, where the experiment was repeated at least three times.

resistance[61,62] (Supplementary Fig. 11). FOSL1 has been shown to play a key role in the crosstalk between MEK and Hippo pathways. Misregulation of MEK pathway in driving tumor growth is well established[63], and key Hippo pathway members (YAP, TAZ) have been implicated in NSCLC[64]. Pham et al.[61] showed that a significant decrease in FOSL1 expression was observed when YAP1-amplified cells were treated with a combination of YAP1 knockdown and Cobimetinib (MEKi), but not for either treatment. BCL6 plays a key role in mediating core cell functions such as antiapoptosis and DNA Damage recognition and has been shown to play a key role in NSCLC[65]. Tran et al.[62] showed that inhibition of BCL6 in NSCLC cell lines conferred sensitivity to Gefitinib. Further experiments also showed that targeting BCL6 and EGFR as a combination showed significant synergy.

NRF2 transcription factor and downstream signaling (more specifically the Keap1-Nrf2 pathway disregulation) has also been implicated in driving EGFR TKI resistance in lung cancer[66]. In addition to identifying KEAP1, the recommendation system also identified CAND1 as a marker of osimertinib resistance. Increased CAND1 expression has been recently shown to be implciated in NSCLC tissues[67]. Altogether these observations indicate that our recommendation approach suggested not only well known resistance markers, but was also able to identify previously unexplored and promising drivers of resistance.

## Discussion

In this study, we explored how an open-ended biological question—discovering drivers of drug resistance in lung cancer—can be approached as a recommendation problem. The current protocol to find resistance markers starts with high-throughput CRISPR screens, followed by a lengthy and time-consuming manual triage, which requires specialized knowledge. Our goal was to replace the manual triage with a recommendation solution that outputs genes potentially driving the resistance. Undoubtedly, such a biomedical setting is not a traditional area of application for recommendation systems. Still, the need to find a small number of relevant genes

amongst prohibitively large set of possibilities constitutes a typical recommendation task. How to approach it?

Common recommendation problems are often solved with collaborative filtering[68], content-based[69] or hybrid solutions[70]. The idea behind collaborative filtering is to predict preferences of a user for an item based on weighted preferences of other users[68]. This approach was demonstrated to work successfully to recommend conceptually complex entities, such as movies, without any additional information about the entity itself[71]. However, to work accurately collaborative filtering requires either large datasets of users actively rating items or a substantial amount of historical data. Owing to the lack of user interactions data, or its equivalent, collaborative filtering is not directly applicable to the CRISPR hit re-ranking task.

On the other hand, content-based approaches recommend items exclusively based on the properties of an item and do not require user interaction data. Therefore, this class of methods seems more suitable for our case. Such content-based recommender would require a set of features, describing properties of a gene relevant for secondary resistance in lung cancer. The disadvantage of content-based approaches is their reliance on similarities between items to make a recommendation. They require either a point of reference or more broadly - training data to make predictions. In the CRISPR hit re-ranking case training data is lacking mainly because a clear conceptual representation of a "good resistance hit" is also lacking. The closest analog of training data in this case would be a weakly labeled dataset[72] created based on information about genes implicated in drug resistance in related types of cancer under similar treatments. It still remains to be proved, however, if a notion of resistance is directly transferable between very specific cancer contexts. Moreover, in this study we focus on discovery drug resistance markers that have never been explored before. Hence it won't make sense to base recommendations on a handful of genes that are already known to be implicated in drug resistance.

In short, CRISPR hit re-ranking task does not immediately fit into classic recommendation frameworks. Complexity of the

biological context, lack of insight about mechanisms of resistance, coupled with absence of training data and user feedback make it difficult to build a fully fledged recommendation solution capable of reasoning about drug resistance.

Devoid of user feedback and training data we approached CRISPR re-ranking problem as a decision-making task. We formalized the problem as a multi-objective optimization with the goal to find solutions defined by optimal trade-offs between novelty, druggability, clinical and pre-clinical relevance. This approach operates in an unsupervised domain, where a perfect solution is unknown. Still we can leverage domain expertise to loosely define a profile of a "good resistance hit". This can be formalized by including/excluding objectives, setting directions of optimization and setting constraints on the objectives themselves.

Similar to the manual triage, re-ranking of CRISPR hits through multi-objective optimization can be performed iteratively. By experimenting with diverse sets of objectives one could explore plausible solutions given the constraints of underlying data. We recognized such iterative decision-making could benefit from an interactive interface. Therefore we developed a Shiny app, SkywalkR, that accounts for diversity of opinions and allows the users to explore the resistance space more efficiently. The app allows the experts to compose custom optimization preferences and re-rank the list of hits based on them. The overall technical solution includes: (1) an automated pipeline that generates features; (2) an interactive interface that runs on top of multi-objective optimization model and has access to the features. This set-up helped to dramatically reduce the time required to prioritize CRISPR screen hits from the usual weeks-months to a few minutes.

The key component of our recommendation system is a hybrid set of features, which is tailored to the EGFR inhibitor resistance context. Along with the usual types of evidence used by the experts during the manual triage, such as clinical and pre-clinical evidence, we included features derived from a custom heterogeneous biomedical knowledge graph. The rationale behind it is that structural properties of the knowledge graph can express relevance of a gene, even if a direct proof of its association with resistance does not yet exist in the literature or clinical/pre-clinical sources. In other words graph-derived features could aid discovery of less-explored or unexpected drivers of drug resistance.

To determine if our recommendation approach produces meaningful results we adopted a hybrid validation strategy. Recommendations were initially evaluated by domain experts, followed by targeted experimental validation of a few promising genes. The in silico evaluation demonstrated that majority (86%) of suggested genes were classified by independent experts as credible and/or novel. The remaining genes (14%) were classified by the experts as "previously unknown hits with unclear tractability", meaning at the time path to biological validation was not obvious. To complement the expert evaluation we picked six genes from the "known" and "previously unknown" categories (WWTR1, MET, PTEN, NF1, KCTD5, EZH2 and SRC) for experimental validation. We demonstrated that deregulation of these genes by genetic and pharmacological means indeed mediated stable resistance phenotype. Though the exact mechanism of this effect remains to be investigated, our experiments proved that recommended genes are implicated in secondary resistance. In summary, hybrid evaluation demonstrates that our recommendation approach not only produces relevant results, but also does it in a fraction of a time compared with the manual triage.

Though the results are promising, we recognize that overall our recommendation approach has a few limitations and areas of improvement. First of all, when applying the multi-objective optimization approach to the CRISPR hit triage problem, there is a risk to obtain unbalanced solutions in some cases. Such solutions occupy margins of a Pareto front and can result from some genes having relatively high values according to just a single objective. Secondly, there is a risk to consider an excessive number of objectives/lines of evidence. The more objectives we take into account, the broader and more topologically complex Pareto fronts can become. This effect limits our ability to unambiguously select a small set of optimal solutions. A few strategies can help to overcome this problem: (1) select a small number of the most important objectives relying on the domain knowledge; (2) multiple objectives can be combined into a single one using scalarization techniques[73]; (3) introduce adaptive weights to individual objectives based on the domain knowledge and the notion of the relative importance of each type of evidence[36]; (4) multi-objective optimization can be performed in consecutive stages on a sub-selection of objectives, similar to Markov decision process[74]. Some of the above approaches could be combined, for example scalarization and adaptive weights. Worth mentioning that all of the listed strategies rely on domain knowledge. Perhaps, they can be viewed as a progressive way to translate individual researcher bias into a formal model.

Another limitation we faced in this study is difficulties with assessing the accuracy of our recommendations. Primarily it stems from the lack of clear notion of a "good resistance hit". One way to address this would be to rely on clinical significance estimates[50] to define reference hits. However, such approach would down-prioritize previously unexplored targets due to lack of clinical data support.

Finally, unlike in traditional recommendation settings, we could not rely on user feedback to gradually evaluate and improve predictions. The ultimate source of truth in our case is experimental validation, where a role of a gene in driving resistance phenotype could be tested in vitro or in animal models. At the moment large-scale validation experiments are not feasible since they are costly and take long time to perform. However in an ideal scenario, experimental output can become an equivalent of user feedback and be used to improve predictions of biomedical recommendation systems.

In summary, accumulation of large amounts of biomedical data coupled with the need to comprehend and reason about it makes biomedical applications an attractive field for recommendation techniques. However, direct translation of traditional recommendation approaches to the biomedical domain is not always trivial. Specifics of the problem space and complexity of biological systems call for efficient recommendation solutions that could operate in unsupervised or weakly supervised settings. We believe that wider adoption and systematic use of recommendation system to solve biological problems bear a potential to transform biomedical research and drug discovery.

## Methods

**EGFRi CRISPR screen design**. Genome-wide CRISPR knockout and activation screens were performed in the EGFR mutant cell lines PC-9 and HCC827 (exon 19 EGFR deletions), as well as gefitinib-resistant clones harboring the secondary EGFR T790M resistance mutation (PC-9T790M and HCC827T790M). The cells were treated with the EGFR inhibitors gefitinib and osimertinib to model resistance to EGFR inhibitors in the 1st and 2nd line clinical settings[38].

For genome-wide loss-of-function CRISPR screens ("CRISPRn'"), cell lines were transduced with a sgRNA library targeting 18,010 human genes. For genome-wide gain-of-function (activation) CRISPR screens ("CRISPRa"), cell lines were transduced with a sgRNA library targeting 23,430 coding isoforms with a unique transcription start site. After selection, library transduced cells were treated with either gefitinib or osimertinib (100 nM each) over 21 days to select for resistance genes. The MAGeCK[75] algorithm was used to identify genes significantly enriched in treatment vs. control arms across all six studies.

**Analysis of CRISPR-pooled screens**. The quality of the sequencing data was assessed using fastqc[76] and mutltiqc[77]. The guide sequences were mapped to the Kosuke Yusa 3 library with custom scripts. The resulting raw counts were quality

controlled in terms of read depth per sample, diversity of guide RNA within a sample using Gini coefficients and in terms of expected clustering of the samples.

Three main comparisons were studied (i) Control samples vs. Treated samples named CvT, (ii) Control samples vs. Plasmid samples named CvP, and (iii) Treated samples vs. Plasmid sample named TvP. The CvT differential analysis was performed with MAGeCK v0.5.7[75]. The CvP and TvP differential analyses were performed with CRISPRCleanR[78] followed by BAGEL[79]. The threshold of significance for BAGEL essentiality results was calculated as described in ref. [80] and for a selected false-discovery rate (FDR) of 5%. The CvP comparison analyses were used for further quality assessments, i.e., identification of core essential genes according to the list published in ref. [81] and expression level of the essential genes. This latest assessment aimed at highlighting any potential false positives by verifying that genes called as essential were expressed in the basal expression profiles of the cell lines. The basal expression data used was published in refs. [80,82].

The three comparisons CvT, CvP and TvP were then aggregated using desirability curves[83] to inform the ranking of genes of interest leading to resistance or to sensitivity. The desirability of a given gene is comprised between 0 and 1. A value of 1 denotes the most interesting genes according to the defined parameters. This score is the aggregation of several partial desirabilities assessing different factors. It enables for instance to take into account for both p-values and log fold-change (LFC) at the same time when ranking the genes of interest. For resistance, the following parameters were considered:

- (i) the partial desirability assessing the FDR of CvT is set to 0 if the FDR is higher than 0.1 and to 1 if the FDR is lower. This enables to focus on the significant genes only.
- (ii) the partial desirability assessing the p-value of CvT is set to 0.01 when the p-value is higher than 0.1. As p-values gets lower, it increases rapidly to reach 1 for a p-value of $10^{-4}$. This rule enables to give a higher desirability to genes with a lower p-value.
- (iii) the partial desirability assessing the size effect of CvT is set to 1 when the effect size is higher the mean log fold-change (LFC) + 3 x standard deviation of the LFC and drops sharply to 0.01 if the LFC is lower than this boundary. This rule enables to give a higher desirability to genes with a higher size effect. Above three times the standard deviation, the desirability plateaus to a maximum of 1.
- (iv) to remove any essential list of genes of interest, the partial desirability assessing essentiality of a gene is set to 0 if it was called as significantly essential in CvP and CvT. If genes with negative LFC remain, their partial desirability is also put to 0. This rule filters out a number of false positives arising because, when knocking done slow essential genes, cells in the treated arm might be dying less quickly than the cells in the control arm leading to a false enrichment in CvP.

For sensitivity, the partial desirabilities (i) and (ii) described above were used. Moreover, the partial desirability assessing the size effect of CvT is set to 1 when the effect size is lower the mean log fold-change (LFC) – 3 x standard deviation of the LFC and drops sharply to 0.01 if the LFC is higher than this boundary.

The partial desirabilities were aggregated as described in ref. [83] and used to rank the genes of interest to be used for further analysis.

Key terms:

- TvC—Treated samples vs. Control samples
- CvP—Control samples vs. Plasmid samples
- OverallDesi—Overall Desirability
- posFDR—Positive false-discovery rate
- TvCCvP_OverallDesi—Aggregated TvC and CvP comparison with Desirability scores
- TvC_posFDR—Aggregated TvC with posFDR to capture statistical significance.

**CRISPR Screen QC and essentiality assessment**. As part of our CRISPR screen processing framework, we developed a pipeline that performs a final QC check to ensure that the screen is of good quality and comparable to other screens performed within our functional genomics center. This analysis assesses whether the expected essential genes can be predicted from the screen. We compared our results with Hart et al.[80,81], and Cancer Dependency Map[82].

We expected and observed the AUC dropout for these essential gene list comparisons to be >0.9 (Hart et al.[81]: 0.91, Behan et al.[80]: 0.99, DepMap[82]: (0.96). Further, we also measure the log fold-change (LFC) distribution in control vs. plasmid to confirm that the LFC distribution is centered around 0, which highlights that there are no survival issues with the cell population under examination (Supplementary Fig. 12).

**CRISPR screen-derived features**. In total we identified a starting list of >3000 resistance genes as hits to be re-ranked. A summary consistency metric was defined as a total number of cell lines, where a gene came up as a hit. Hits were defined according to desirability scores. We used two different types of thresholds to define hits for CRISPRn and CRISPRa screens: (*TvCCvP_OverallDesi* > 0.7 and *TvC_posFDR* < 0.1) and (*TvCCvP_OverallDesi* > 0.7 and *TvC_posFDR* < 0.1),

respectively. We considered consistency in CRISPRn, CRISPRa screens separately, as well as the overall consistency in the screen (CRISPRa and CRISPRn combined).

**Graph-derived features**. The biomedical knowledge graph was built as described in[32]. Embeddings were calculated based on the full graph using RESCAL algorithm[84]. L2 distance from human gene nodes to "EGFR" and "NSCLC" nodes was calculated using Faiss package[85]. Full graph was also used to calculate descriptive network metrics such as node degree and number of unique neighbors connected to a node.

To make graph-derived metrics more relevant to mechanistic explanation of EGFRi resistance, we further focused on a protein–protein interaction (PPI) subgraph. PPI subgraph was defined based on the "interacts" edges from HetioNet[86,87]. PPI subgraph was used to calculate PageRank[39] and betweenness[88] metrics for each gene node.

We included Jupyter notebooks, illustrating how graph-derived features can be generated on biomedical knowledge graphs (https://github.com/AstraZeneca/skywalkR-graph-features)[89]. Owing to licensing restrictions we are not able to share our knowledge graph, however similar work can be performed on open access heterogeneous biomedical graphs such as Hetionet[86,87].

**Literature-based features**. To estimate overall literature support for a gene's involvement in EGFR inhibitor resistance we analyzed a corpus of PubMed and PMC articles as well as Springer bio-classified data published between 2000 and 2019. The corpus was further restricted to a set of 185,299 publications relevant to cancer and/or secondary drug resistance. The search was performed on the title, abstract and full text. Next, two key terms were identified—"EGFR" and "NSCLC". For each of the target terms we computed the total number of papers that mention a given human gene and one of the key terms of interest together. To account for the fact that total number of published papers per gene differs greatly, we have included normalized literature frequencies. The resulting four summary metrics were included in a hybrid feature set and exposed through the SkywalkR interface.

In addition to single-gene metrics we computed multi-term gene co-occurrence in previously defined (cancer and drug resistance) context. The idea behind this analysis was to discover sets of genes that tend to co-occur together in EGFR inhibitor resistance context. Frequent co-occurrence of gene combinations across publications could indicate a strong link between genes within a set and suggest a potential mechanism driving secondary resistance. Gene co-occurrence matrix was used to build an interactive heatmap in the SkywalkR app.

**Clinical enrichment features**. Osimertinib is an irreversible EGFR inhibitor that selectively targets the EGFR T790M mutation[90]. We aggregated data from osimertinib-treated patients across five clinical trials—AURAext, AURA2, AURA3, FLAURA, and ORCHARD. AURAext is a phase II extention of the AURA trial with an 80 mg/day osimertinib dose administered to non-small cell lung cancer (NSCLC) patients[41]. AURA2 is a Phase 2 single arm clinical trial for NSCLC patients with advanced disease who progressed on previous treatment with EGFR Tyrosine Kinase Inhibitor (TKI), and also carry the EGFR T790M mutation[42]. AURA3 is a Phase 3 randomized study comparing the efficacy of osimertinib vs. platinum-based chemotherapy in advanced NSCLC patients who have progressed on prior treatment with EGFR TKI. These patients also carry the EGFR T790M mutation. FLAURA is a Phase 3 clinical trial for first-line osimertinib treated advanced NSCLC patients vs. other EGFR TKI standard-of-care (SoC) treatments[27]. ORCHARD is a Phase 2 platform study in patients with advanced NSCLC who have progressed on first-line osimertinib treatment[43].

In all, 335 patients treated with osimertinib across the trials were analyzed. Sequencing data from Guardant Health and FMI gene panels were utilized to identify genetic alterations. Patients who were classified with the RECIST criteria of Partial Responder/Complete Responder AND PFS > 6 mos were classified as responders, and enrichment of genetic alterations in responders vs. non-responders was calculated. As individual trials utilized different clinical gene panels, the enrichment metrics were kept trial-specific and no aggregation across trials was performed. This "enrichment score" was used as a feature for multi-objective optimization.

**Tractability**. To estimate druggability of candidate genes we relied on tractability scores from OpenTargets https://github.com/melschneider/tractability_pipeline_v2,[44]. Tractability scores for three modalities were included: antibodies, small molecules and other modalities (e.g., enzyme, protein, oligonucleotide, etc.). For convenience buckets were reversed, so that the highest druggable bucket corresponded to the highest numeric score 10. Reversed tractability metrics were exposed through the SkywalkR app.

**Gene essentiality**. Information about whether the gene is common essential or not was retrieved from DMC DepMap[82]. We flagged a gene as "essential" if inhibition of target gene results in reduced viability for 90% of cell lines used in DepMap. Otherwise a gene was flagged as "nonessential".

**Transcriptomic features**. The features "RNAseq_LFC" and "RNAseq_pval" that were included in the recommendation system framework resulted from an

unpublished internal experiment to capture the effect of acute Osimertinib treatment on gene upregulation within the cell lines. The experiment involved treating the cell lines PC9 and HCC827 with Osimertinib and DMSO, respectively, to capture transcriptomic changes pre- and post-treatment with Osimertinib. The features derived from the experiment have been included as part of our public code and data repository https://github.com/AstraZeneca/skywalkR,[46]. Raw RNAseq data is available under the accession number GSE193259.

**Models and implementations**. Pareto fronts were computed based on the implementation from the rPref R package[91]. During the multi-objective optimization we were looking for solutions that occupy an optimal surface (front), which represents the best trade-offs between the considered objectives. This optimal surface is labeled as Pareto level 1. Subsequently, sub-optimal surfaces by definition contain worse (less optimal) solutions and are labeled Pareto level 2, ... n. The SkywalkR interface returns genes that occupy Pareto level 1. Solutions can be further sorted within a Pareto front by the users by one or more variables of choice.

To build binary classifiers for gene labels we used fast implementation of random forest from ranger package[92]. Shapley values were calculated using fastshap package[93]. Correlation plot was generated with corrplot package[94].

**Pathway enrichment analysis**. We used MetaCore (Clarivate, https://portal.genego.com/) to perform enrichment analysis. Genes were matched to possible targets in functional ontologies of MetaCore. The probability of a random intersection between a set of IDs the size of target list with ontology entities is estimated in *p*-value of hypergeometric intersection. The lower *p*-value means higher relevance of the entity to the dataset. Canonical pathway maps represent a set of signaling and metabolic maps for human. All maps are created by manual curators/scientists from Clarivate Analytics relying on published peer-reviewed literature. To adjust enrichment results we additionally performed crosstalk analysis.

**Clinical significance of recommended hits**. To assess clinical relevance of hits recommended by our framework, we compared our list to MSK's FDA-approved Precision Oncology Knowledge base OncoKB[50]. OncoKB assesses genetic alterations by annotating each gene under five categories: therapeutic, prognostic, diagnostic, resistance, and FDA-levels). Specific definitions can be found in OncoKB[50].

**Generation of KO or activation cell lines, plasmids and antibodies**. PC-9, II-18, HCC4006, NCI-H1975 and HCC827 cell lines were cultured at 37 °C and 5% $CO_2$ in RPMI 1640 GlutaMAX media (Gibco, US) supplemented with 10% fetal bovine serum. For generation of KO cell line pools PC-9, HCC827 and Il-18 cells were transduced with pKLV2-EF1a-Cas9Bsd-W (Addgene ID:68343)[95] to stably express Cas9. Transduced cells were subjected to Blasticidin (Gibco, US) selection. To knockout target genes guide RNAs targeting KCTD5, NF1, PTEN, EZH2 or non-targeting-control (see Supplementary Table 4) were cloned into pLKV2-U6gRNA5(BbsI)-PGKpuro2ABFP-W (Addgene ID:67974) and Cas9-expressing cell lines were transduced and selected with puromycin (Gibco, US). To activate MET expression in PC-9 cells the three vector-based CRISPR SAM system[96] was used. In brief, viroid's containing open reading frames of dCas9-VP64 and MS2-P65-HSF1 as well as target specific guideRNA expression constructs were purchased (SAMVP64BSTV, SAMMS3HYGV and LV06, respectively, Sigma-Aldrich) and used to stepwise transduce and select PC-9 cells. After each transduction cells were subjected to respective antibiotic selection (blasticidin, hygromycin, puromycin). Fourteen days after transduction whole-cell lysates of selected cell pools were analyzed by western blot to confirm CRISPR KO or CRISPR activation of gene expression using anti-EZH2 (CST, #5246, 1:1000) anti-KCTD5 (proteintech, 15553-1-AP, 1:1000), anti-MET (CST, #8198, 1:1000), anti-NF1(abcam, ab17963, 1:1000), anti-PTEN (CST, #9559, 1:1000), anti-WWTR1(CST, #70148, 1:1000) as well as loading control antibodies anti-Tubulin (Sigma-Aldrich, T9026, 1:5000) or anti-Vinculin (Abcam, ab18058, 1:5000). Unprocessed western blot scans are included as supplementary data in this manuscript.

**Viability assay**. Two-thousand cells were seeded 24 h prior to treatment into 96-well plates. Cells were treated with indicated concentrations of DMSO, osimertinib or ECF-506 and cultured for 4 or 5 days as technical triplicates. For the Osimertinib + Tazemetostat combination, cells were pre-treated with Tazemetostat for 3 days, followed by treatment with Osimertinib + Tazemetostat for 6 days at the indicated concentrations. Considering that this experiment was to confirm the KO readout, it was performed once with three technical replicates (Fig. 6C). Viability was determined by adding CellTiter-Glo Luminescent Cell Viability Assay (Promega, US) according to manufacturer's protocol and measuring luminescence using a SpectraMax plate reader (Molecular Devices, US). Results are visualized as percent viability of DMSO treated control. Shown are representative results of three biological replicates.

**Flow cytometry-based resistance tracking**. For studying the effects of gene KO on osimertinib resistance, PC-9, II-18 and HCC827 KO cell lines were co-cultured

with respective Cas9-expressing control cells and treated with DMSO or osimertinib (50 nM) for 14 days. Media and drug was replenished every 3–4 days. Fractions of KO cells relative to control cells were determined by measuring BFP co-expression in KO cells via flow cytometry at indicated time points. In case of PC-9 CRISPR activation lines, PC-9 control cells were stably labeled with BFP and fractions of unstained CRISPR activation cells relative to BFP-positive control cells were determined by flow cytometry. All experiments were performed as three technical replicates of two independent guideRNAs per gene as biological replicates. FlowJo 10 was used for analysis of FACS data.

**Clonogenic assay**. PC-9 cells were seeded in 6-well plates and 24 h later treated with indicated concentrations of osimertinib or DMSO for 21 days or 10 days, respectively. Media and treatment was replaced every 3 to 4 days. After indicated treatment periods cells were PBS washed, fixed (BD Cytofix, BD, US) and stained with 0.01% (w/v) crystal violet (Sigma-Aldrich,US). Plates were scanned on a GelCount plate scanner (Oxford Optronics, UK). All experiments were performed as two technical replicates of two independent guideRNAs per gene as biological replicates.

**Acquired-resistant cell lines**. Osimertinib-resistant cell lines were derived from sensitive parental lines as previously described[60]. Briefly, cells were plated in individual 14 mm wells and cultured in growth media containing either low-dose (10 nM) osimertinib, which was progressively increased to a high-dose treatment (500 nM) over a 6-week time frame, or treated initially with high-dose osimertinib. Cells avidly proliferating in high-dose osimertinib were further expanded in media containing 160 nM osimertinib, banked, and used for subsequent experiments.

**Reporting summary**. Further information on research design is available in the Nature Research Reporting Summary linked to this article.

## Data availability
All data generated, analyzed, and interpreted have been included as part of the code repository https://github.com/AstraZeneca/skywalkR[46]. Specifics have been included in the Methods section. For clinical data features, aggregated enrichment scores have been included, but not individual patient-level data as these are proprietary at the time of drafting the manuscript. However, these trials have been published and appropriately referenced in the manuscript. We utilized two features from an internal RNAseq study that we described briefly in the additional methods document. The data itself has been included in our code repository for reviewers and readers to access. The raw RNAseq data used in this study is available in the GEO database under accession code GSE193259. The Knowledge Graph as it was used for this study cannot be released due to proprietary data included in it. However, the construction of the KG was as described in Geleta et al.[32].

## Code availability
SkywalkR source code and documentation can be found at https://github.com/AstraZeneca/skywalkR[46]. Notebooks, illustrating how to generate graph-derived features based on a biomedical knowledge graph can be found at https://github.com/AstraZeneca/skywalkR-graph-features[89].

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

## Acknowledgements

We thank David Geleta, Andriy Nikolov, Gavin Edwards, and Benedek Rozemberczki for technical and scientific support. We thank Steven Criscione, Aleksandra Markovets, Ryan Hartmeier and Aisha Swaih for expert feedback on candidate hits, and Daniel Barrell for his insights into CRISPR screen QC. We thank the Early Computational Oncology and Functional Genomics Center teams for scientific input and feedback.

## Author contributions

K.B., U.M., B.S., J.D. and E.P. conceived the study. A.G. designed and implemented the computational framework and analyzed the data. D.P., V.P., M.U., M.A. and K.B. generated and analyzed the data. M.P. performed the CRISPR screen and A.B. implemented the QC pipeline. M.P., M.M. and H.T. carried out wet lab experiments and analyzed the data. A.G. drafted the manuscript in consultation with K.B., D.P., E.P., M.P. and M.A. All authors reviewed the manuscript.

## Competing interests

The authors declare the following competing interests. All authors except M.P. and H.T. were full-time employees and shareholders of AstraZeneca at the time of study. M.P. and H.T. were PostDoc Fellows of the AstraZeneca PostDoc program at the time when experiments were completed. J.D.'s current affiliation is Tempus Labs, Cambridge MA. E.P.'s current affiliation is DeepMind, London, UK.
