## [Peer Review File · Nature Communications]

Reviewers' Comments:

Reviewer #1:

Remarks to the Author:

The manuscript successfully built a hybrid recommendation system to find key drivers of resistance in EGFR mutant NSCLC. Some suggestions are listed as below.

1. Page 4, sentence "a summary metric reflecting overall consistency in the full screen", please illustrate specifically the method to compute the "summary metric".
2. In ranking process of CRISPR screen data, have you considered the effect strength of each gene? As declared in Table 1, the seven features generated from CRISPR screen data are all about number of cell lines where same gene is a hit. In this way, the scores of negative selection (or positive selection) computed by original CRISPR data are ignored. For example, supposing gene A knocked-out in cell line X generates a $\log(\text{score negative selection})$ of - 6.0 while gene B knocked-out in cell line X gets a $\log(\text{score negative selection})$ of - 2.0, they all count for 1 point in "KO_osi". But clearly, gene A should rank higher than gene B.
3. About clinical enrichment features, what's the meaning of "internal clinical study 1", "internal clinical study 2" and "cross-studies"?
4. Page 6, sentence "The resulting list contained 36 recommended genes". What's the specific criteria to screen out the 36 genes?
5. Page 7, sentence "both sensitive to osimertinib (Figure 7, A)", it should be (Figure 5, A).
6. Page 7, the sentence "The assay showed that after 14 days of co-culture perturbation of NF1, PTEN or KCTD5 expression did not affect proliferation compared to control cells" isn't consistent with the results of Figure 5B.
7. Page 7, most of the genes chosen for experimental validation belongs to "known" category. It may be more meaningful to validate genes in "novel, credible" and "novel, not credible" categories experimentally.
8. Page 9, there are errors in the sentence " Though the exact mechanism of this is effect remains to be investigated".
9. In this study, the recommendation system was built to find key genes driving drug resistance of EGFR inhibitors in EGFR mutant non-small cell lung cancer. As there are many similar biological problems, it would be beneficial to many researchers if the authors could provide a standardized program for similar biological explorations to compute knowledge graph-based features. Strategies to overcome EGFR TKI resistance have been well studied, especially for the genes validated in this study. Therefore, the relevance of the approach established in this study was strongly recommended to be validated by other regimens such as Herceptin resistance in breast cancer.
10. Page 2, the abbreviation "TKI" was not defined at the initial use.

Reviewer #2:

Remarks to the Author:

Key results & potential significance

The presented work evaluates the relevance of applying a recommender system strategy to support EGFR TKI resistance gene prioritization. The main hypothesis is that a combination of clinical data, literature co-occurrence counts, and metrics from the knowledge graph structural analysis, can unveil relevant recommendations to speed up the screening. A multi-objective optimization framework has been chosen to produce recommendations from Pareto fronts. Relevant recommendations have been obtained, including known and a posteriori verified resistance markers. Evaluation has been conducted using A/B testing by domain experts, pathway enrichment analysis, and experimental validation. Interpretation of feature impact on relevance has been proposed using Shapley values for classification model explanation. In addition, an open-source implementation of an interactive interface to generate recommendations (skywalkR) has been proposed.

This work is original and tackles an important challenge in non-small cell lung cancer treatment, a research focus with high societal impact. The proposed method can undoubtedly be applied to a large range of other biomedical questions, assessing a common shortcoming inherent to the large-scale nature of current biological research. Expanding from the few proof-of-concept works cited in

this manuscript, the authors made a convincing case for the adaptation of recommender system methods to biomedical research, which shows great potential in my opinion. The challenges associated with such endeavour are particularly well defined in this manuscript and could help further research. Regarding the methodology aspects, the current approach to biomedical RS is, in my opinion, original and clever. The novelty and significance of the recommendations are, however, beyond my expertise.

Validity of the conclusion & quality of the methodology

Regarding the validity of the conclusion, as stated in the discussion and introduction, validation of the recommendations is a difficult enterprise due to the absence of a 'gold standard'. The over-representation of EGFR TKI resistance pathways membership among the recommendations suggest an 'overall' relevance of the results set. The individual recommendation assessment has been done through expert evaluation of 36 recommendations and experimental validation of 4 of them, from an initial candidate set of >1000 genes (the exact value in the main document would have been appreciated).

It is not explicitly stated in the manuscript if those 36 constitute the whole Pareto optimal set (which is what we guess from the interface) or if a selection of a subset has been performed, or if a cutoff has been applied to Pareto ranks. A more detailed method section regarding the recommendations and the method applied to define the 'Pareto levels' would be beneficial. Similarly, the manuscript is also lacking an explanation on which basis the 4 have been selected for experimental validation. I think such information is required in the manuscript to assess the validity of the results.

I believe that expert assessment of 'placebo recommendations', i.e. genes from random draws in lower Pareto levels, would have strengthened the claim by the authors, if those would tend to fall into the 'not credible' category more often than true recommendations.

However, the poor value obtained on multiple objectives in the 'not credible' set provide some evidence that the features used for recommendations are relevant (Fig3). It is however surprising that those low-scoring genes end up in the recommended set. Could the authors provide some explanation on why the RNAseq feature is the only one that can yield recommendations on its own?

Finally, it is claimed that the credible recommendations "would potentially have been missed" during manual triage. The recommendation is framed as a "re-ranking" of CRISPR hits, yet there is no description of an initial ranking, on which I assume the manual triage is based. A better description of the common triage criteria would help to grasp in which extent the former claim is valid.

The proposed methodology is sound, even if some feature definitions might be polished. From our understanding, the literature-based features represent the total number of articles mentioning both the gene and one of the key terms 'EGFR' or 'NSCLC' in a cancer context. However, the size of the available literature can be highly heterogeneous between well studied genes (eg. P53) and more rarely discussed genes, potentially inducing bias in the recommendations. Therefore, the normalized frequency could be a better estimate of the strength of the relation in the literature, as well as being directly comparable between genes.

Article clarity

The article is globally well written and comprehensible, but the relations between the evaluation results and the feature description is a bit confusing, and somewhat can even be misleading. The title suggests a "Knowledge Graph-based Recommendation Framework", the first figure depicts graph metrics (including clustering coefficient not mentioned in the objectives), and the results section's preamble contains detailed description of these features. However, from my understanding, the KG based features were not included by default and thus did not drive the evaluated recommendations, nor helped finding the novel markers (but, from the Shapley values, could have). If it is the case, the impact of the extra objectives exclusion/inclusion on the

recommendations should be discussed more*, if it is not, some part in the results section should be clarified.

*to go further, it could be interesting to compare the Pareto front obtained with and without the extra features to show their contribution, rather than indirectly through their predictive power regarding expert-assessed credibility, which, to me, seems a rather convoluted analytical choice.

Regarding clarity, the word "hybrid" is frequently used to characterize the heterogeneous feature set and by extension the proposed method that relies on them. However, the word 'hybrid' is quite often used to describe a recommender system that leverages both content-based and collaborative filtering, which is not what the proposed RS is. We suggest re-wording to avoid confusion.

The context is well described, although mention of "gene prioritization" prior works is lacking. While not framed as a recommendations problem and not commonly based on the methodology presented here, they aim for a similar goal and should be of interest for the readers.

Authors' Response to Reviews of

Knowledge Graph-based Recommendation Framework Identifies Novel Drivers of Resistance in EGFR mutant Non-small Cell Lung Cancer

Anna Gogleva, Dimitris Polychronopoulos, Matthias Pfeifer, Vladimir Poroshin, Michaël Ughetto, Matthew Martin, Hannah Thorpe, Paul Smith, Ben Sidders, Miika Ahdesmäki, Ultan McDermott, Eliseo Papa*, Krishna Bulusu*.

Nature Communications,

RC: *Reviewers' Comment*, AR: Authors' Response, □ Manuscript Text

We thank the reviewers for their comments and taking time to review our work. We have integrated reviewer's suggestions in the revised version of our manuscript. The most significant changes include:

- we revised a set of default objectives and included graph-derived features, PageRank and betweenness. New recommended list included 21 additional genes which were validated by five independent experts. Majority of the recommended genes (49, 86%) were labelled by the experts as either *known* or *novel and credible*;
- we performed additional experimental validation for three additional recommended genes from the 'novel and credible' category as indicated by reviewer1 and editor – WWTR1, EZH2 and SRC. We collected experimental evidence, indicating that modulation of their expression affects osimertinib resistance;
- we included Jupyter notebooks illustrating how graph-derived features can be computed on a heterogeneous biomedical knowledge graph. We believe these notebooks together with the open-sourced skywalkR code would be sufficient to extend our recommendation approach to similar biological problems.

Please find our point-by-point responses to all the reviewer's comments below.

1. Reviewer #1, Remarks to the Author

The manuscript successfully built a hybrid recommendation system to find key drivers of resistance in EGFR mutant NSCLC. Some suggestions are listed as below.

1.1. Page 4, sentence "a summary metric reflecting overall consistency in the full screen".

RC: *Please illustrate specifically the method to compute the "summary metric".*

AR: We have included further details as to the definition of consistency in the *Methods section*. The consistency metric was defined as a total number of cell lines, where a gene was a hit. Hits were defined according to desirability scores *Supplementary Methods (Analysis of CRISPR pooled screens)*. We included additional details in *Methods* section to clarify this point. We considered consistency in CRISPRn, CRISPRa screens separately, as well as overall consistency (CRISPRa and CRISPRn combined).

1.2. Ranking process

RC: *In ranking process of CRISPR screen data, have you considered the effect strength of each gene? As declared in Table 1, the seven features generated from CRISPR screen data are all about number of cell lines where same gene is a hit. In this way, the scores of negative selection (or positive selection) computed by original CRISPR data are ignored. For example, supposing gene A knocked-out in cell line X generates a log(score negative selection) of - 6.0 while gene B knocked-out in cell line X gets a log(score negative selection) of - 2.0, they all count for 1 point in “KO_osi”. But clearly, gene A should rank higher than gene B.*

AR: We agree that effect size would have an impact on the ranking. However, in this work we decided that the most meaningful features extracted from CRISPR screens would be the ones based on desirability scores, which binary define if a gene is a hit or not. We argue that consistency of a gene’s performance across conditions is a more informative metric to predict the final relevance of a gene, rather than the effect sizes in the individual CRISPR screens.

1.3. Clinical studies

RC: *About clinical enrichment features, what’s the meaning of “internal clinical study 1”, “internal clinical study 2” and “cross-studies”?*

AR: These scores refer to internal clinical studies. We renamed the variables to make them more descriptive and updated the feature definitions indicating which clinical trials were included for these features (Please see *Supplementary Table 1*). In a nutshell, *ES_1* refers to enrichment score calculated based on AURAext, AURA2, and AURA3 clinical studies; *ES_2* refers to enrichment score calculated based on FLAURA clinical study; *ES_3* refers to enrichment score refers to a comparative account of FLAURA with ORCHARD study. The relevant part of Methods section has also been updated with the necessary references.

1.4. Page 6, sentence “The resulting list contained 36 recommended genes”.

RC: *What’s the specific criteria to screen out the 36 genes?*

AR: The reason to further focus on the 36 genes was because this set was obtained by largely mimicking decision-making process that domain scientist go through during manual triage. This is just one of the lists that can be generated with our approach using skywalkR interface (each combination of objectives likely to result in a different set of genes). Since it was not feasible to experimentally validate all lists, we focused on the one that made sense to the domain experts. In the current version of the manuscript we included 2 more graph features in the set of objectives, this resulted in 21 additional genes, which we subsequently validated (Please see updated *Figure 3*).

1.5. Page 7, sentence “both sensitive to osimertinib (Figure 7, A)”

RC: *It should be (Figure 5, A)*

AR: We thank the reviewer for noticing the discrepancy. We have corrected the figure reference.

1.6. Page 7, the sentence “The assay showed that after 14 days of co-culture perturbation of NF1, PTEN or KCTD5 expression did not affect proliferation compared to control cells”

RC: *isn’t consistent with the results of Figure 5B.*

AR: We have updated the figure caption to clarify that **Figure 5B** is a schematic, explaining the principle of the flow-cytometry based resistance tracking experiment. The actual experimental data for NF1, PTEN and KCTD5 is shown on the panel *C*.

1.7. Page 7

RC: *Most of the genes chosen for experimental validation belongs to “known” category. It may be more meaningful to validate genes in “novel, credible” and “novel, not credible” categories experimentally.*

AR: We agree with the reviewer and have now addressed this with further experimental validation studies for hits in 'novel and credible' category. We would also like to highlight our thought process behind the choice of hits for these experiments. For experimental validation of our recommendations, we identified biological mechanisms that have recently gained prominence within the domain of EGFRi resistance but do not yet have an approved drug target within the pathway (Hippo pathway and KCTD family). In addition, we selected targets that have inhibitors available to assess combination benefit in resistance models when a recommended target is inhibited in combination with osimertinib. Lastly, to provide ground truth we included established EGFRi resistance markers as the background to our validation studies.

We are happy to report that in the interim we have managed to produce additional experimental validation for WWTR1, SRC and EZH2. Please see updated **Results, Figures 5-6, Supplementary Figures 9 and 10.**

1.8. Page 9

RC: *There are errors in the sentence “Though the exact mechanism of this is effect remains to be investigated”*

AR: We thank the reviewer for pointing this out. We have rephrased the sentence.

1.9. General comment

RC: *In this study, the recommendation system was built to find key genes driving drug resistance of EGFR inhibitors in EGFR mutant non-small cell lung cancer. As there are many similar biological problems, it would be beneficial to many researchers if the authors could provide a standardized program for similar biological explorations to compute knowledge graph-based features. Strategies to overcome EGFR TKI resistance have been well studied, especially for the genes validated in this study. Therefore, the relevance of the approach established in this study was strongly recommended to be validated by other regimens such as Herceptin resistance in breast cancer.*

AR: Indeed, we believe our approach can be easily extended to a wide range of similar problems, mostly by swapping the feature set we used with context-specific features. We have successfully applied it to several internal projects, including breast cancer. Our approach yielded promising targets that undergo expert review. Unfortunately we are not able to provide more details at this stage as these studies are still at an early stage and we wanted the primary message of this publication to focus on NSCLC. However, following the reviewer's suggestion we have added a repository Jupyter notebook (<https://github.com/AstraZeneca/skywalkR-graph-features>) that illustrates how to generate graph-derived features based on a reasonably complex biomedical knowledge graph (Hetionet, <https://het.io/>). We believe, this notebook together with already open-sourced interactive apps (skywalkR, repository <https://github.com/AstraZeneca/skywalkR>) is sufficient for the interested practitioners to extend our approach to similar problems, such as Herceptin resistance in breast cancer.

1.10. Page 2

RC: *The abbreviation “TKI” was not defined at the initial use.*

AR: We added a definition for the 'TKI' abbreviation.

2. Reviewer #2, Remarks to the Author

2.1. Key results & potential significance

RC: *The presented work evaluates the relevance of applying a recommender system strategy to support EGFR TKI resistance gene prioritization. The main hypothesis is that a combination of clinical data, literature co-occurrence counts, and metrics from the knowledge graph structural analysis, can unveil relevant recommendations to speed up the screening. A multi-objective optimization framework has been chosen to produce recommendations from Pareto fronts. Relevant recommendations have been obtained, including known and a posteriori verified resistance markers. Evaluation has been conducted using A/B testing by domain experts, pathway enrichment analysis, and experimental validation. Interpretation of feature impact on relevance has been proposed using Shapley values for classification model explanation. In addition, an open-source implementation of an interactive interface to generate recommendations (sky-walkR) has been proposed.*

RC: *This work is original and tackles an important challenge in non-small cell lung cancer treatment, a research focus with high societal impact. The proposed method can undoubtedly be applied to a large range of other biomedical questions, assessing a common shortcoming inherent to the large-scale nature of current biological research. Expanding from the few proof-of-concept works cited in this manuscript, the authors made a convincing case for the adaptation of recommender system methods to biomedical research, which shows great potential in my opinion. The challenges associated with such endeavour are particularly well defined in this manuscript and could help further research. Regarding the methodology aspects, the current approach to biomedical RS is, in my opinion, original and clever. The novelty and significance of the recommendations are, however, beyond my expertise.*

AR: We thank Reviewer #2 for their positive comments and appreciation of our approach. We believe recommender system have a lot of potential in helping scientists to build and navigate complex hypotheses.

2.2. Validity of the conclusion quality of the methodology

RC: *Regarding the validity of the conclusion, as stated in the discussion and introduction, validation of the recommendations is a difficult enterprise due to the absence of a ‘gold standard’. The over-representation of EGFR TKI resistance pathways membership among the recommendations suggest an ‘overall’ relevance of the results set. The individual recommendation assessment has been done through expert evaluation of 36 recommendations and experimental validation of 4 of them, from an initial candidate set of >1000 genes (the exact value in the main document would have been appreciated).*

AR: We revised the text and added additional details (Please see **Page 4**). The total number of initial candidate genes was 1550.

RC: *It is not explicitly stated in the manuscript if those 36 constitute the whole Pareto optimal set (which is what we guess from the interface) or if a selection of a subset has been performed, or if a cutoff has been applied to Pareto ranks. A more detailed method section regarding the recommendations and the method*

applied to define the 'Pareto levels' would be beneficial.

AR: We thank the reviewer for highlighting this lack of clarity. We have now added details to the **Methods** section to clarify how top genes are selected (Please see **Page 13, Models and implementations**). During the multi-objective optimization we were looking for solutions that occupy an optimal surface (front) which represents the best trade-offs between the considered objectives. This optimal surface is labelled as *Pareto_level_1*. Subsequently, sub-optimal surfaces by definition contain worse (less optimal) solutions and are labelled *Pareto_level_2, ... n*. The skywalkR interface returns genes that occupy *Pareto_level_1*, with no additional filters applied. Solutions can be further sorted by a user's variable of choice.

RC: *Similarly, the manuscript is also lacking an explanation on which basis the 4 have been selected for experimental validation. I think such information is required in the manuscript to assess the validity of the results.*

AR: We have now included an explanation for the basis of selecting the genes for experimental validation. We wanted to address 3 primary aspects - 1) identify biological mechanisms that have recently gained prominence within the domain of EGFRi resistance but do not yet have an approved drug target within the pathway (Hippo pathway and KCTD family). 2) In addition, we selected targets that have inhibitors available to assess combination benefit in resistance models when a recommended target is inhibited in combination with osimertinib. 3) To provide ground truth we included established EGFRi resistance markers as the background to our validation studies. (Please see updated **Results, Methods, Figures 5-6, Supplementary Figures 9-10**).

RC: *I believe that expert assessment of 'placebo recommendations', i.e. genes from random draws in lower Pareto levels, would have strengthened the claim by the authors, if those would tend to fall into the 'not credible' category more often than true recommendations.*

AR: We agree with the reviewer, it could be interesting to make a formal comparison between solutions residing on different Pareto levels. However, by definition all lower Pareto levels already contain less optimal or sub-optimal solutions. Since full-scale evaluation by the experts is very time and resource intense, we were not able to perform it for complete lists to make a meaningful assessment. We do however would like to mention that the reason behind having five expert reviewers independently assessing the credibility of these hits was to capture if frequency with which each hit is annotated as credible/not-credible. We still believe that the ultimate validation is experimental evidence. Hence, we have included additional experimental evidence for WWTR1, SRC and EZH2 genes, supporting their involvement in osimertinib resistance (Please see updated **Figures 5-6, Supplementary Figures 9-10** and respective **Results** and **Methods** sections).

RC: *However, the poor value obtained on multiple objectives in the 'not credible' set provide some evidence that the features used for recommendations are relevant (Figure 3). It is however surprising that those low-scoring genes end up in the recommended set. Could the authors provide some explanation on why the RNAseq feature is the only one that can yield recommendations on its own?*

AR: This is an interesting point, to address it we have updated **Discussion** section, when talking about limitations of our approach. A potential explanation here is twofold. Firstly, with our multi-objective optimization approach we sometimes can end up with solutions on the margins of Pareto fronts, e.g having relatively high values according to just a single objective. The effect might be damped because some objectives are partially correlated (we have added additional **Supplementary Figure 7**). Secondly, expert assessments of recommendations can be subjective, in a sense that the experts tend to value objectives differently. In some cases, a strong signal in relevant RNAseq assays, combined with lack of other types of evidence and very few mentions in the literature can be sufficient to investigate the potential target further, so the target is added to 'novel, but credible category'.

RC: *Finally, it is claimed that the credible recommendations "would potentially have been missed" during manual triage. The recommendation is framed as a "re-ranking" of CRISPR hits, yet there is no description of an initial ranking, on which I assume the manual triage is based. A better description of the common triage criteria would help to grasp in which extent the former claim is valid.*

AR: We agree with the reviewer that it is hard to assess if the recommendations would have been missed indeed by the manual process, so we have deleted the specific sentence, and have also included a description of the 'initial ranking' which is essentially a statistical threshold driven workflow that resulted in 1550 significant hits. (Please See updated **Page 3**).

RC: *The proposed methodology is sound, even if some feature definitions might be polished. From our understanding, the literature-based features represent the total number of articles mentioning both the gene and one of the key terms 'EGFR' or 'NSCLC' in a cancer context. However, the size of the available literature can be highly heterogeneous between well studied genes (eg. P53) and more rarely discussed genes, potentially inducing bias in the recommendations. Therefore, the normalized frequency could be a better estimate of the strength of the relation in the literature, as well as being directly comparable between genes.*

AR: We thank the reviewer for this suggestion. We have included normalized frequencies in the feature set. We have updated data and widgets in the skywalkR app, and also revised **Methods** section and **Supplementary Table 1**.

2.3. Article clarity

RC: *The article is globally well written and comprehensible, but the relations between the evaluation results and the feature description is a bit confusing, and somewhat can even be misleading. The title suggests a "Knowledge Graph-based Recommendation Framework", the first figure depicts graph metrics (including clustering coefficient not mentioned in the objectives), and the results section's preamble contains detailed description of these features. However, from my understanding, the KG based features were not included by default and thus did not drive the evaluated recommendations, nor helped finding the novel markers (but, from the Shapley values, could have). If it is the case, the impact of the extra objectives exclusion/inclusion on the recommendations should be discussed more*, if it is not, some part in the results section should be clarified.*

AR: We agree with the reviewer that validation in the current version of the manuscript is incomplete. We believe that graph-derived features can offer additional insights when recommending targets, especially in the absence of clinical or pre-clinical features. However, graph-derived features are not typically considered by the experts when performing manual triage, which is what our study aims to address. We have now included graph features in the default set, subsequent changes are listed below:

- We have included PageRank and betweenness to the list of default objectives and generated a new list of recommendations, which contains 57 genes (Please see **Figure 3** and updated **Results** section).
- We also run it through the expert validation, and additionally experimentally validated WWTR1, SRC and EZH2 genes (Please see **Figures 5-6, Supplementary figures 8-9**).
- Updated Shapley analysis (Please see **Figure 4**) confirms that graph-derived features closely follow CRISPR-derived features in the classification task.

Altogether these findings suggest that graph-derived features in combination with relevant context-specific

features can be valuable for target triage.

RC: **to go further, it could be interesting to compare the Pareto front obtained with and without the extra features to show their contribution, rather than indirectly through their predictive power regarding expert-assessed credibility, which, to me, seems a rather convoluted analytical choice.*

AR: Indeed, different sets of objectives result in Pareto fronts of different shape and size. To illustrate this we performed an ablation experiment. We systematically excluded a single category of features at a time from a default set of features used to obtain the list of 57 genes (default recommendations). We then calculated Jaccard index between the ablated version and the default list. The results are listed in the table below:

Feature Category	Jaccard index	Pareto front size
literature	0.9122807	52
crispr	0.6842105	39
clinical	0.7368421	42
pre-clinical	0.5614035	32
graph	0.6315789	36

RC: *Regarding clarity, the word "hybrid" is frequently used to characterize the heterogeneous feature set and by extension the proposed method that relies on them. However, the word 'hybrid' is quite often used to describe a recommender system that leverages both content-based and collaborative filtering, which is not what the proposed RS is. We suggest re-wording to avoid confusion.*

AR: We thank the reviewer for the valuable suggestion. We have adjusted the text throughout of the manuscript and replaced the instances of 'hybrid recommendation system' by just 'recommendation system' when referring to our approach.

RC: *The context is well described, although mention of "gene prioritization" prior works is lacking. While not framed as a recommendations problem and not commonly based on the methodology presented here, they aim for a similar goal and should be of interest for the readers.*

AR: We have updated the Introduction and have referenced a few prior papers, offering solutions to gene prioritization problem (Please see *Introduction, Page 2*).

Reviewers' Comments:

Reviewer #1:

Remarks to the Author:

I have no further comments. It could be accepted for publication now.

Reviewer #2:

Remarks to the Author:

The proposed manuscript has been improved since last revision. The authors responded to all of our concerns and took note of our suggestions, for which we are grateful.

All the discussed features, notably the graph measures, have been included by default in the objectives, which yield new recommendations that the authors took time to thoroughly evaluate, adding extra experimental validation along the way that strengthen the main claim. We do not have further concerns regarding this manuscript clarity and the validity of the underlying work. Please find below some final comments and optional suggestions for improvements.

The authors also provided us with results of exclusion experiment, for which we are very thankful. They show how each feature category shapes the final recommendations, which complements the Shapley value experiment. Although I believe a full upset-plot of this simpler analysis would have been a nice addition to this article, I admit that the Shapley Values, by characterizing the selection of the most interesting recommendations by expert, might be of better interest to the readers.

Regarding the impact of graph measure, I think that a brief description of the kind of entities and relationships represented by nodes and edges would be beneficial, since the KG construction is expected to greatly impact those values. It is also not clear to me if the KG is available and under which license. In order to support further work, maybe this could be precised, or the authors could point other biomedical open-KG that could be used to support similar work.

In the authors' response is mentioned an updated description of the initial ranking in page 3, which I could not find.

We thank the authors for the manuscript and the interesting discussions in the response.

Reviewer #3:

Remarks to the Author:

The manuscript by Gogleva et al., elegantly tackles a series of challenges in cancer and beyond, most notably how and which genes mediate response and resistance to both general and targeted therapies, by establishing a multidisciplinary framework that incorporates both empirical and computational data. As such, the manuscript will be of widespread interest and impactful to the fields of cancer and beyond. Notably, the authors employ sophisticated computational analyses and algorithms, including recommender systems that have been recently successfully exploited for high dimensional analyses of CRISPR screens (e.g. PMID: 31395745 and others). As such, the analytical framework is at the forefront and will be of wide interest to the community.

I have two major comments:

1. How feasible will the method be when incorporating widely different CRISPR screening datasets that can have dramatically different dynamic ranges (and thereby variable confidence in true hit-calling)? For instance, PMID: 32514112 shows that different screens can vary widely in how they call true hits based on the quality and dynamic range of the screens performed. This can lead to both false positives and false negatives, and I noticed that this concern was also brought up by Reviewer #1. The statement by the authors is not fully convincing in this regard, so what are the different stringent sets of criteria used to incorporate or not a given screening data set? This needs to be discussed more explicitly so that other less experienced members of the community can incorporate these methods into their own work.

2. The authors mention incorporating the advice of experts to curate hits, but I think it would be important to also integrate algorithms that have also proven to have predictive power in the clinic, particularly as they relate to druggable factors. For instance, the authors could easily compare/contrast their hit-lists with OncoKB (<https://www.oncokb.org/>) to highlight readily available druggability vs not and how that might inform future usage of their methods.

Still, overall, this is a good manuscript that is well-written and well-presented. Lastly, I applaud the authors for their transparency in terms of data analyses (e.g. Jupyter notebooks). Thumbs up from this

NatComms, response to reviewers, 2nd round

REVIEWER COMMENTS

Reviewer #1 (Remarks to the Author):

I have no further comments. It could be accepted for publication now.

We thank the reviewer #1 for taking time to review our work and their insightful feedback.

Reviewer #2 (Remarks to the Author):

The proposed manuscript has been improved since last revision. The authors responded to all of our concerns and took note of our suggestions, for which we are grateful.

All the discussed features, notably the graph measures, have been included by default in the objectives, which yield new recommendations that the authors took time to thoroughly evaluate, adding extra experimental validation along the way that strengthen the main claim. We do not have further concerns regarding this manuscript clarity and the validity of the underlying work. Please find below some final comments and optional suggestions for improvements.

The authors also provided us with results of exclusion experiment, for which we are very thankful. They show how each feature category shapes the final recommendations, which complements the Shapley value experiment. Although I believe a full upset-plot of this simpler analysis would have been a nice addition to this article, I admit that the Shapley Values, by characterizing the selection of the most interesting recommendations by expert, might be of better interest to the readers.

We thank the reviewer for their positive comments on our work and appreciate their time taken to review the revised version of our manuscript.

Regarding the impact of graph measure, I think that a brief description of the kind of entities and relationships represented by nodes and edges would be beneficial, since the KG construction is expected to greatly impact those values. It is also not clear to me if the KG is available and under which license. In order to support further work, maybe this could be precised, or the authors could point other biomedical open-KG that could be used to support similar work.

We agree with the reviewer #2 that graph composition has a strong impact on graph-derived features. The procedure of our graph construction, schema, as well as datasets included is described in detail in a separate manuscript (Geleta et al, 2021, <https://www.biorxiv.org/content/10.1101/2021.10.28.466262v1.abstract>, now accepted at WSDM MLog Workshop). Unfortunately, we are not able to share the complete graph at this

moment due to certain licencing restrictions and proprietary datasets included. Similar work can be supported to open access heterogeneous biomedical graphs such as Hetionet (<https://het.io/>) using the Jupyter notebooks we provided and used for our analysis. Please see updated *Methods* section (*Graph-derived features*).

In the authors' response is mentioned an updated description of the initial ranking in page 3, which I could not find.

We thank the reviewer for spotting this. Indeed, the reference we provided was not correct. The description of the initial ranking of CRISPR hits can be found in the *Methods* section (*'Pooled drug CRISPR screen data'*).

We thank the authors for the manuscript and the interesting discussions in the response.

We appreciate the thoughtful comments and the time reviewer #2 has taken to consider our work.

Reviewer #3 (Remarks to the Author):

The manuscript by Gogleva et al., elegantly tackles a series of challenges in cancer and beyond, most notably how and which genes mediate response and resistance to both general and targeted therapies, by establishing a multidisciplinary framework that incorporates both empirical and computational data. As such, the manuscript will be of widespread interest and impactful to the fields of cancer and beyond. Notably, the authors employ sophisticated computational analyses and algorithms, including recommender systems that have been recently successfully exploited for high dimensional analyses of CRISPR screens (e.g. PMID: 31395745 and others). As such, the analytical framework is at the forefront and will be of wide interest to the community.

I have two major comments:

1. How feasible will the method be when incorporating widely different CRISPR screening datasets that can have dramatically different dynamic ranges (and thereby variable confidence in true hit-calling)? For instance, PMID: 32514112 shows that different screens can vary widely in how they call true hits based on the quality and dynamic range of the screens performed. This can lead to both false positives and false negatives, and I noticed that this concern was also brought up by Reviewer #1. The statement by the authors is not fully convincing in this regard, so what are the different stringent sets of criteria used to incorporate or not a given screening data set? This needs to be discussed more explicitly so that other less experienced members of the community can incorporate these methods into their own work.

We thank the reviewer for rightly pointing out a key aspect of both the CRISPR screens and the resulting downstream analysis. One of the aims of our framework is to reduce the probability of returning false positives. We believe we achieved this by not only relying on

CRISPR-derived features to triage the list of high-value hits, but by incorporating features derived from the knowledge graph, clinical, preclinical and literature-based features (see Results, ***A hybrid set of features supports recommendation system***).

We have expanded the ***Methods*** section and included additional details on analysis and QC of CRISPR data (Please see Methods, ***EGFRi CRISPR screen design, CRISPR Screen QC and essentiality assessment, Analysis of CRISPR pooled screens, Supplementary figure 12***). We believe this should be sufficient to guide other scientific practitioners through the process of analysing CRISPR data and extracting features for recommender systems.

Whilst we recognise that false positive rate can never be zero, from the CRISPR analysis point of view we followed stringent measures to reduce the risk:

1. ***CRISPR screen design***: We used CRISPR data after QC by the experimental team as an input for the knowledge-graph based recommender system. We also used libraries with at least 3 guides per gene, and no guide was excluded during the analysis stage. QC of CRISPR data was performed according to industry standards, including biological replicates to identify outliers. (Please see Methods, ***CRISPR Screen QC and essentiality assessment***).
2. ***Essentiality***: We control for core essentiality of the genes, and compare ‘hit’ retrieval with three established essential gene lists (Please see Methods, ***CRISPR Screen QC and essentiality assessment***).
3. ***Desirability***: We implemented a statistical methodology to assess the quality of CRISPR hits. We calculated a ‘desirability score’ to assess statistical significance of a hit that takes into consideration multiple metrics. This score along with False Discovery Rate (FDR) is used to implement the first set of strict thresholds that feeds into our framework (Please see Methods, ***Analysis of CRISPR pooled screens***).

An interesting question from the reviewers’ comment is if we want to identify ‘high-value’ hits across diverse and independent CRISPR screens. There is a possibility and value in assessing guide performance over multiple screens, and rank accordingly via the desirability score – something that cannot be done on the basis of one screen (as in our study), this would definitely improve identification of false negative/positives in the future. In reference to our framework, each of these screens will need to go through their individual QC pipeline based on the study design before entering our Essentiality and Desirability workflows as described above.

In addition to QC measures applied to the CRISPR screen data, our recommendation approach takes into accounts different types of evidence. Even though we observed that CRISPR features significantly drive recommendations, graph features contribute the next highest (please see Results, ***Shapley values indicate the high impact of CRISPR-derived features*** Figure 4). We believe this inclusive approach helps to derive a more balanced set of recommendations.

Finally, the relationship and knowledge-exchange between the results of our framework and the EGFR biology domain experts is a critical part of identifying true hits. For complete transparency, we asked our experts to independently tag what they considered as ‘not

credible' (Please see Figure 3) which further adds to identifying False Positives as defined by expert knowledge as opposed to computational metrics.

2. The authors mention incorporating the advice of experts to curate hits, but I think it would be important to also integrate algorithms that have also proven to have predictive power in the clinic, particularly as they relate to druggable factors. For instance, the authors could easily compare/contrast their hit-lists with OncoKB (<https://www.oncokb.org/>) to highlight readily available druggability vs not and how that might inform future usage of their methods.

We thank the reviewer for raising the aspect of clinical translatability of these results, which ultimately is the goal for all preclinical studies. Our framework already includes druggability (referred to as 'tractability') as part of the hybrid feature set (see Supplementary Table 1). In the interactive interface this feature can be included or excluded from the optimization task. The users also have an option to select from tractability with small molecules, antibodies or other modalities, such as enzymes.

Regarding the aspect of using patient data, we already include features from Osimertinib clinical trials to assess significance of the recommended hits (see Methods, **Clinical enrichment features**). As per the reviewer's suggestion, we have now also annotated our recommended list of genes with OncoKB, with each 'level' of significance as indicated by Therapeutic, Diagnostic, Prognostic, Resistance, and FDA (Please see **Supplementary Table 3**). We have also indicated in Discussion that using such clinical factors, we can be more precise in terms of translating our results to the clinic but will run the risk of losing out on truly novel markers that as of now do not have any clinical significance evidence.

Still, overall, this is a good manuscript that is well-written and well-presented. Lastly, I applaud the authors for their transparency in terms of data analyses (e.g. Jupyter notebooks). Thumbs up from this Reviewer.

We thank the reviewer #3 for taking their time to review our work, and for acknowledging the transparency we aimed to achieve with our study.

Reviewers' Comments:

Reviewer #3:

Remarks to the Author:

The authors have addressed all of my concerns. In fact, I believe they went above and beyond while ensuring maintenance of rigor & reproducibility, which is much appreciated. Congratulations on your study.

Reviewers' Comments and Author Response

Reviewer #3 (Remarks to the Author):

The authors have addressed all of my concerns. In fact, I believe they went above and beyond while ensuring maintenance of rigor & reproducibility, which is much appreciated. Congratulations on your study.

AR: We thank the reviewer for their time in assessing our study, and for valuable and kind feedback.